

# Vertical distributions of aerosol optical properties during the spring 2016 ARIAs airborne campaign in the North China Plain

Fei Wang[1,2], Zhanqing Li[1,3], Xinrong Ren[3,4,5], Qi Jiang[6], Hao He[3], Russell R. Dickerson[3], Xiaobo Dong[7], Feng Lv[7]

[1]State Key Laboratory of Earth Surface Processes and Resource Ecology and College of Global Change and Earth System Science, Beijing Normal University, Beijing, 100875, China
[2]Key Laboratory for Cloud Physics, Chinese Academy of Meteorological Sciences, Beijing, 100081, China
[3]Department of Atmospheric and Oceanic Science, University of Maryland, College Park, MD 20742, USA
[4]Air Resources Laboratory, National Oceanic and Atmospheric Administration, College Park, MD, USA
[5]Cooperative Institute for Climate and Satellites, University of Maryland, College Park, MD, USA
[6]National Meteorological Center, Beijing, 100081, China
[7]Hebei Provincial Weather Modification Office, Shijiazhuang, 050021, China

*Correspondence to*: Zhanqing Li (zli@atmos.umd.edu)

**Abstract.** Vertical distributions of aerosol optical properties derived from measurements made during 11 aircraft flights over the North China Plain (NCP) in May-June 2016 during the Air Chemistry Research In Asia (ARIAs) were analyzed. Aerosol optical data from in situ aircraft measurements shows good correlation with ground-based measurements. The regional variability of aerosol optical profiles such as aerosol scattering and backscattering, absorption, extinction, single scattering albedo (SSA), and the Ångström exponent (α) are for the first time thoroughly characterized over the NCP. The SSA at 550 nm showed a regional mean value of 0.85±0.02 with moderate to strong absorption and the α ranged from 0.49 to 2.53 (median 1.53) indicating both mineral dust and accumulation mode aerosols. Most of the aerosol particles were located in the lowest 2 km of the atmosphere. We describe three typical planetary boundary layer (PBL) scenarios and associated transport pathways as well as the correlation between aerosol scattering coefficients and relative humidity (RH). Aerosol scattering coefficients decreased slowly with height in the clean PBL condition, but decreased sharply above the PBL under polluted conditions, which showed a strong correlation ($R^2 \geq 0.78$) with ambient RH. Back-trajectory analysis shows that clean air masses generally originated from the distant north-western part of China while most of the polluted air masses were from the heavily polluted interior and coastal areas near the campaign area.

## 1 Introduction

Aerosol loading in the East China are exceptionally heavy and highly variable due to drastically increases in the emissions of pollutants from anthropogenic activities in the region during the last several decades. Changes in air quality and climate are strongly coupled (Li et al., 2016), and both have tremendous impact on public health of the dense population in the region (Kan et al., 2012).



Aerosol particles are considered to be an important radiative forcing agent in the climate system but the detailed effects have been most uncertain (Stocker et al., 2013). Aerosols modify the local and regional climate by absorbing and scattering solar radiation, and the radiation budget through the aerosol direct effect (Charlson and Hofmann, 1992). Aerosols can also affect cloud-precipitation processes and aerosol-cryosphere interactions through indirect and semi-direct effects (Twomey, 1974;

Lohmann and Feichter, 2005; Andreae and Rosenfeld, 2008; Nair et al., 2013). Several recent studies have shown that the magnitude of precipitation is strongly correlated to aerosol concentration (Zhao et al., 2006; Li et al., 2011; Tao et al., 2012), through various mechanisms as summarized most recently in Li et al. (2017a). Precipitation frequency and intensity are also altered by the long-term impacts of aerosols (Guo et al., 2016; 2017). Much of the uncertainty in aerosol radiative forcing comes from the variability of optical properties in anthropogenic aerosol such as scattering, absorption, Ångström exponent

($\alpha$), and aerosol optical depth (AOD). These properties depend strongly on particle size distribution, chemical composition, and the ambient relative humidity (Anderson et al., 2003). The spatial and temporal variations of these properties, especially horizontal and vertical distributions, are essential factors in the effects of aerosols on both climate and the environment (Haywood and Boucher, 2000).

Aerosol scattering and absorption coefficients ($\sigma_{sca}$ and $\sigma_{abs}$, respectively) are important parameters related to atmospheric

characteristics such as visibility and air quality. Observed $\sigma_{sca}$ and $\sigma_{abs}$ are generally distinct within and above the PBL. The structure of the PBL, in part determined by the vertical distribution of the aerosol extinction coefficient ($\sigma_{ext}$, associated with $\sigma_{sca}$ and $\sigma_{abs}$), is a crucial factor in estimating aerosol pollution conditions (Yu et al., 2002; Dong et al., 2017), partially due to their strong interactions/coupling (Li et al., 2017b). In addition, given the variability of PBL structure and its interaction with aerosols, the transport of pollutants is yet to be fully understood. For example, dust aerosols could be lifted into the free

troposphere, i.e., above the PBL, and transported over a long distance, altering aerosol vertical distributions over remote areas (Han et al., 2008). Topographically-generated local circulations can carry high concentrations of surface air pollutants and change the PBL structure (Chen et al., 2009). Such variable aerosol vertical distributions can alter the optical properties of aerosols such as AOD, thus affecting the regional radiation balance (Liu et al., 2012). Distinguishing between local air pollutants and pollutants transported from other source regions, and identifying transport patterns under different atmospheric

circulation conditions also need to be addressed.

The three-dimensional information about aerosol optical properties (especially vertical distributions) is of importance but such measurements are scanty, and in need of expansion (Kahn et al., 2017). Given the potential impact of aerosols on climate, accurate measurements of the vertical profiles of aerosol optical properties are needed. These data can be obtained directly or indirectly from platforms such as meteorological towers (Zhao et al., 2017), tethered balloons (Stratmann et al., 2003; Ferrero

et al., 2010), and unmanned aerial vehicles (Corrigan et al., 2008). Airborne and surface-based remote sensing are two common approaches to obtain direct and indirect aerosol observations. Although limited in terms of temporal and geographic coverage, airborne sensing can provide direct, high resolution, and in situ aerosol vertical profiles, used to evaluate numerical models and satellite retrievals. Lastly, aircraft in situ measurements are also an appropriate and complementary tool to use in obtaining aerosol optical properties (Chazette and Liousse, 2001).



Airborne instruments have been used to characterize aerosol properties in the lower troposphere around the world (Ulla et al., 2002; Haywood et al., 2003a; Taubman et al., 2004; Taubman et al., 2006; Hains et al., 2008; Ferrero et al., 2011; Ryder et al., 2013; Kim et al., 2015; Schwarz et al., 2016; Babu et al., 2016), as well as in China, e.g., in Beijing (Zhang et al., 2006; Liu et al., 2009; Zhang et al., 2009; Zhang et al., 2011), Dongbei (Dickerson et al., 2007), Hebei (Sun et al., 2013), and Shanxi (Li

et al., 2015a; 2015b). Most of the measurements were made for parameters such as aerosol number concentration or size distribution.

In May-June 2016, a comprehensive joint ground and airborne experiment was conducted in the north China Plain (NCP). This study aimed to examine the consistency of airborne and surface-based measurements, and to evaluate aerosol radiative characteristics and the distribution and transport of air pollutants both horizontally and vertically. It is an integral part of, also

a foundation for, a series studies on aerosol-cloud-climate interactions in a densely populated and yet fast developing region of China (Li et al., 2017a).

A twin-engines turboprop airplane was deployed to measure trace gases and aerosol optical properties in the lower atmosphere, simultaneously observed using ground-based in situ and remote measurements in Xingtai supersite. The airplane flew ascents and descents in the boundary layer and the lower free troposphere to obtain vertical profiles of aerosol optical properties. High-

15 resolution aircraft measurements of aerosol optical properties give detailed information about the occurrence, extent, and evolution of aerosol vertical distribution. Aerosol optical properties such as $\sigma_{sca}$ and $\sigma_{abs}$, backscattering, extinction, SSA, $\alpha$, and AOD from the airborne measurements are presented and discussed.

Details about the field experiment and the instruments used are given in Section 2. The vertical and regional distributions of aerosol optical properties are presented in Section 3. In Section 4, the structure of the PBL under both clean and polluted

conditions, the correlation between vertical $\sigma_{sca}$ and relative humidity (RH), and back trajectory analyses are discussed. Section 5 summarizes the major conclusions from this study.

## 2 Experimental Description

### 2.1 Sites and flight information

The Aerosol Atmosphere Boundary-layer Cloud (A$^2$BC) campaign took place in Hebei Province of the NCP, about 300 km

south of Beijing, from May to December 2016. An intensive observation period (IOP) was from May to June when airborne observations were conducted using two airplanes to measure aerosol and cloud properties, respectively. A supersite site located in Xingtai (XT; 114.36°E, 37.18°N, 182 m above sea level, or ASL) was established. The NFS also sponsored the project "Air chemistry Research In Asia" (ARIAs). NASA's Korean US Experiment (KORUS-AQ) was conducted at roughly the same time. One of the airplanes (Y-12) that measured aerosol properties was based at the Luancheng Airport (LC; 114.59°E, 37.91°N,

58 m ASL), located in southeast of Shijiazhuang, the capital of Hebei Province. The airplane was flown to three locations in the area to conduct spirals from ~300 m to ~3.5 km, as shown in Fig. 1. These locations include XT, Julu (JL; 115.02°E,



37.22°N, 30 m ASL), and Quzhou (QZ; 114.96°E, 36.76°N, 40 m ASL). All four sites are located to the east of the Taihang Mountains with XT right at the foothill of the mountain range. A total of 11 flights were conducted during the A$^2$BC IOP period (Table 1).

Comprehensive measurements of aerosol optical properties were made during the field campaign using the instrumented turboprop Y-12 airplane operated by the Weather Modification Office of Hebei Meteorological Bureau. The typical speed of the aircraft is 60-70 m s$^{-1}$ with ascent/descent rates of 2-5 m s$^{-1}$. The aircraft was equipped with multiple aerosol and gas measurement instruments, rigorously tested and calibrated during a ground-based campaign to optimize instrument performance. Table 2 summarizes the major instruments deployed on the plane and ground.

### 2.2 Aircraft Instruments

#### 2.2.1 Nephelometer

Measurements of $\sigma_{sca}$ from the Y-12 aircraft were made using an integrating Nephelometer (TSI, model 3565) at three wavelengths: blue (450 nm), green (550 nm), and red (700 nm). Details about the instrument calibration and uncertainties have been described elsewhere (Anderson and Ogren, 1998; Anderson et al., 2009). The Nephelometer was calibrated prior to the field campaign using $CO_2$ gas and filtered zero air as described in the instrument manual. It aspirates air at a flow rate of 30 L min$^{-1}$ through a common air inlet. Data were recorded at a frequency of 1 Hz.

The actual range of total scattering angles captured by the nephelometer is less than the ideal range of 0° to 180°. To correct the biases, we adjusted with empirically-derived angular truncation correction factors using the calculated α (Anderson and Ogren, 1998). Due to the hygroscopic growth of aerosols, the scattering values are adjusted using a correction factor f(RH) (Anderson et al., 2003; Shinozuka et al., 2007), which is calculated as:

$$f(RH) = \left[ \frac{(100 - RH_{neph})}{(100 - RH_{amb})} \right]^{\gamma}, \tag{1}$$

$$B_{sca\_adj} = B_{sca} \times C \times f(RH), \tag{2}$$

where $RH_{neph}$ is the internal nephelometer RH, $RH_{amb}$ is the ambient RH, $\gamma$ is an experimentally determined dry versus humid factor of the hygroscopicity (Beyersdorf et al., 2016), $B_{sca\_adj}$ is the adjusted scattering coefficient, $B_{sca}$ is the measured scattering coefficient, and C is the angular truncation factor. During the study period, the relative uncertainty after calibration was around a few percent (evaluated from the reproducibility of laboratory measurements).

#### 2.2.2 PSAP

A particle soot absorption photometer (PSAP, Radiance Research, 565 nm) without interference by scattering signals provides highly sensitive absorption measurements. The measurement principle of the PSAP and errors caused by non-absorbing particulate have been described elsewhere (Bond et al., 1999; Sheridan et al., 2005; Virkkula et al., 2005), and the PSAP has





been used near Beijing (Chaudhry et al., 2007). To ensure a steady sample flow when using the instrument during flights, we monitored the total flow and PSAP flow rates to enhance the accuracy of measurements. The instrument operates at a flow rate of 2 L min$^{-1}$ when one-minute measurement averages are used. The raw absorption coefficients from the PSAP, $\sigma_{abs, \lambda=565nm}$, were corrected to 550 nm using Eq. 3 for loading:

$$\sigma_{abs,\lambda_i} = \sigma_{abs, \lambda=565nm} \times \frac{565}{\lambda_i}, \qquad\qquad (3)$$

where $\lambda_i$ is the i$^{th}$ wavelength. Ideally, for this calculation, the absorption coefficient would also be measured under ambient conditions, but it is somewhat less dependent on humidification than is scattering (Schafer et al., 2014). Therefore, such calibrated measurement was not done during the field campaign.

### 2.2.3 Meteorological instruments

A cloud water inertial probe (CWIP, Rain Dynamics) measuring meteorological parameters such as pressure, temperature (T), and RH was installed on the Y-12 aircraft. It was calibrated prior to the deployment and synchronized with the global positioning system (GPS) time and geolocation of the scientific data was dually calibrated by the GPS and the compass navigation satellite system (CNSS).

### 2.3 Instrumentation at the Xingtai supersite

Besides the aircraft observation, a full suite of instruments that measured aerosol and meteorological parameters was deployed at the Xingtai supersite. Those of concern to this study includes aerosol optical remote sensing observation (CIMEL radiometer), in situ particle light extinction measurement (Cavity Attenuated Phase Shift particle light extinction monitor, CAPS), and radiosondes of meteorological parameters.

### 2.3.1 CAPS

We present field measurements of ambient aerosol extinction coefficients ($\sigma_{ext}$) by using a cavity attenuated phase shift spectroscopy (CAPS) instrument with a time resolution of 1 s at the Xingtai supersite. This technique has advantages such as high sensitivity, cost-efficiency, easy control and a long effective absorption optical path. The CAPS measures the phase shift of a distorted waveform of the modulated light from a light emitting diode (LED) in the sample cell with two high reflectivity mirrors (Sun et al., 2014). It has a demonstrated sensitivity of less than 2 Mm$^{-1}$ for 1 s sampling periods. A method description,
including results from laboratory characterization and field deployment, has been reported previously (Massoli et al., 2010). The CAPS extinction and the $\sigma_{ext}$ measured by the combination of Nephelometer + PSAP showed a good correlation in both laboratory-generated test particles and ambient aerosols (Petzold et al., 2013). The detailed principle of the CAPS technology, optical path adjustment process and calibration method can be found in previous studies (Kebabian et al., 2008; Kebabian and Freedman, 2007).





### 2.3.2 Radiosondes

For comparisons with airborne measurements, radiosondes (Model DFM-09, Graw Radiosonde GmbH & Co. KG, Germany) were launched from the Xingtai supersite during the IOP. The DFM-09 radiosonde is a lightweight weather radiosonde that measures temperature (at a resolution of 0.1°C with an accuracy of 0.2°C), pressure (at a resolution of 0.1 hPa with an accuracy

of 0.5 hPa), RH (at a resolution of 1% with an accuracy of 2%), and wind speed (accuracy = 0.2 m s$^{-1}$) (Navas-Guzmán et al., 2014). RH is measured by a thin-film capacitance sensor and its uncertainties and errors depend on temperature and GPS location (Steinke et al., 2015). Data acquisition and processing were performed at the Graw ground station located at the Xingtai supersite using the Grawmet software.

### 2.3.3 CIMEL automatic Sun-sky radiometer

To obtain an optical characterization of aerosol vertical distribution and to contrastive analysis with airborne measurements, we used a CIMEL radiometer, the standard instrument used in the Aerosol Robotic Network (AERONET). The CIMEL (CE-318) used at the Xingtai AERONET site is a two detector eight channel (340, 380, 440, 500, 675, 870, 940, and 1020 nm) radiometer designed to make automatic tracking and measuring direct solar and sky radiances with a field of view of approximately 1.2°. AOD and SSA are used to compute at each wavelength except for the 940 nm channel, which is used to

retrieve total column water vapor. The measurement uncertainty for field instruments, primarily due to calibration uncertainty, which is spectrally dependent with higher errors in the UV (Eck et al., 1999). The details of the CIMEL radiometer operating principles and network are described by Holben et al. (1998).

### 3 Vertical and regional distributions of aerosol optical properties

### 3.1 Comparison between aircraft and ground-based measurements

Ground-based measurements collected at XT were matched in time and space with airborne measurements over the site. Fig. 2 shows the vertical profiles of RH and T from the sounding done at 05:55 coordinated universal time (UTC, 13:55 local time) on 8 May 2016 and from aircraft measurements made over the site before the radiosonde launch (04:40–05:40 UTC, one descent spiral and two ascent spirals). Both RH and T derived from sounding data and measured from the aircraft follow the same general trends.

For a final evaluation of the performance of the airborne instruments in measuring ambient aerosol characteristics, $\sigma_{ext}$ measured by the nephelometer/PSAP onboard the aircraft and by the ground-based CAPS PM$_{ex}$ during research flight (RF) 6 (RF6; 21 May), RF7 (28 May), RF8 (28 May), and RF11 (11 June) are shown in Fig. 3. CAPS-measured $\sigma_{ext}$ data collected during the spiralling part of the flight track were averaged and all $\sigma_{ext}$ data were adjusted to a wavelength of 550 nm. In general, mean values of surface $\sigma_{ext}$ were ~12% smaller than the corresponding aircraft-measured $\sigma_{ext}$ located at the bottom of the

profiles.



There is a certain comparability of retrieval parameters (e.g., SSA) between ground-based observation and aircraft measurement. For example, the SSA derived from AERONET aerosol monitoring network in Baltimore-Washington, DC region were on average 0.011 lower than the values derived from aircraft profile measurements (Schafer et al., 2014). For aircraft-AERONET (based on Nephelometer + PSAP, which are also used in this study) comparisons with ground-based

AERONET at multiple sites in South Africa, the SSA with a mean difference of 0.01 (RMS: 0.03) in biomass burning aerosol measurements (Leahy et al., 2007). There is also a small difference between aircraft in situ (0.87) and CIMEL (0.85) measurements in SSA on a day with mixed upper-level smoke aerosol layer and low-level dust layer in the Banizoumbou (Johnson et al., 2010). As for this study, retrievals from a ground-based CIMEL automatic Sun-sky radiometer (CE-318) were compared with aircraft in situ measurements. The SSA at 550 nm retrieved from the CIMEL (0.93) agreed well with airborne

measurements (0.94) on 28 May (Table 3). The well-matched vertical profiles and ground-based data present a more complete profile of aerosol optical information in the lower atmosphere, demonstrating that the high credibility and quality of airborne measurements.

### 3.2 Statistics of aerosol optical properties

A statistical summary of aerosol optical properties retrieved from aircraft measurements made over the four target areas (LC,

JL, QZ, and XT) is given in Table 4. Aerosol scattering coefficients ($\sigma_{sca}$) and backscattering coefficients ($\sigma_{bsca}$) are directly obtained from the nephelometer. The truncation correction was applied and the $\sigma_{sca}$ were corrected for the ambient RH. The mean $\sigma_{sca}$ at 550 nm measured at LC, JL, QZ, and XT were 57.10±67.71, 95.57±106.61, 87.00±100.43, and 75.30±84.58 Mm$^{-1}$, respectively, and the mean $\sigma_{bsca}$ in each target area was 7.67±7.6, 9.91±9.19, 11.12±10.51, and 9.21±8.6 Mm$^{-1}$, respectively. Aerosol absorption properties at LC and XT were stronger than those at JL and QZ. Note that each variable has a large standard

deviation, which suggests that there are large variations associated with these aerosol optical properties.

Mean vertical distributions of $\sigma_{sca}$ and $\sigma_{abs}$ at 550 nm derived from aircraft measurements made over the four target areas are shown in rows 3 and 4 of Fig. 4. In general, the values of $\sigma_{sca}$ and $\sigma_{abs}$ decrease with altitude. Peaks in the $\sigma_{sca}$ profiles at certain heights are seen at JL, QZ, and XT. The $\sigma_{sca}$ profile at LC changes little below 3000 m. Compared with the other target areas, $\sigma_{abs}$ at JL have relatively lower values near the ground and gradually decrease with height, which suggests that there were less

light-absorbing and more scattering composition in aerosols in the atmosphere.

Based on measurements of aerosol scattering and absorption coefficients, aerosol extinction coefficient ($\sigma_{ext}$, defined as the sum of $\sigma_{sca}$ and $\sigma_{abs}$) and SSA ($\omega$, defined as the ratio of $\sigma_{sca}$ to $\sigma_{ext}$) were calculated at 550 nm following Eq. 4. The small mismatch in calculating $\omega$ was corrected by linearly extrapolating the $\sigma_{abs}$ values to the wavelengths of the scattering measurements as defined. Both $\sigma_{ext}$ and $\omega$ are considered as primary determinants for the direct radiative effect of aerosols.

The vertical profile of $\sigma_{ext}$ closely follows that of aerosol mass concentration (Kim et al., 2015). SSA was described as an important factor in controlling whether an aerosol layer has a cooling or warming effect; it can also be used for studying the radiative forcing at the top of the atmosphere (Bergstrom and Russell, 1999). In this study, the $\omega_{\_550}$ in the range 0.68-0.99 with a mean value of 0.85 suggests the importance of absorption aerosols.



$$\omega_\lambda = \frac{\sigma_{sca,\lambda}}{\sigma_{ext,\lambda}} = \frac{\sigma_{sca,\lambda}}{\sigma_{sca,\lambda} + \sigma_{abs,\lambda}}. \qquad (4)$$

The $\alpha$ was calculated using Eq. 5 where $\sigma_{sca,\lambda}$ is the scattering coefficient at a given reference wavelength $\lambda$:

$$\alpha(\lambda_2/\lambda_1) = \frac{-\log(\sigma_{sca,\lambda_2}/\sigma_{sca,\lambda_1})}{\log(\lambda_2/\lambda_1)}, \qquad (5)$$

In this study, we used two wavelengths, $\lambda_1 = 450$ nm and $\lambda_2 = 700$ nm.

An analogous expression can be written for the wavelength-dependence of absorption. In general, $\alpha$ is a basic measure of the aerosol size distribution. It is related to the ratio of fine to coarse particles with $\alpha$ less than 1 for coarse-mode aerosol such as dust particles and $\alpha \sim 2.0$ for very fine-mode aerosol such as smoke particles (Hamonou et al., 1999). Vertical profiles of the median values of $\omega_{550}$ and $\alpha$ are shown in rows 5 and 6 of Fig. 4. The profiles were generated by calculating the median value at each altitude layer from all the measured profiles. The variations seen may reflect changes in the origin of aerosol particles

and transport routes (Léon et al., 2009), although the values fall well within the range of the standard deviations.

Angular-corrected data from the nephelometer is the scattered light intensity in the backward hemisphere of the particle (90°–180°). The backscattering fraction, $\beta_{sca}$, is the ratio of the backscattering coefficient over the total scattering coefficient at a given wavelength (Garland et al., 2009). The ratio of forward scattering to backscattering varies with the particle size parameter and reflects the angular characteristics of particle scattering and the proportion of fine particles (diameters < 2 μm). During

the field campaign, $\beta_{sca}$ remained at a low value below 2000 m Mean values of $\beta_{sca}$ measured in the four target areas were $0.13\pm0.003$, $0.11\pm0.005$, $0.12\pm0.003$, and $0.12\pm0.006$. Fluctuations above 2000 m suggest that particle sizes changed quickly due presumably to different air masses transported from different directions.

Fig. 5 shows the mean AOD at 550 nm for different altitude ranges. The regional mean AOD in each altitude range (< 1 km, 1–2 km, and 2–3 km) was $0.10\pm0.08$, $0.10\pm0.07$, and $0.03\pm0.03$, respectively. Standard deviations are greater than 50% of the

mean, suggesting that AODs varied greatly in the same altitude range. The magnitude of AOD generally decreased with altitude. The effect of the transport of atmospheric pollutants is evident at QZ where the largest AOD values were found in the 1–2 km layer instead of at the surface.

## 4 PBL structure and aerosol $\sigma_{sca}$ as a function of RH

The vertical distribution of aerosol particles is important for determining radiative effects, especially in the presence of clouds.

During the haze episodes, the vertical profiles of CO, the aerosol concentration, and the scattering coefficient were found to be well correlated (Haywood et al., 2003b). Formenti et al. (2003a; 2003b) presented a full analysis of the correlations among these variables. Examples of typical vertical profiles of aerosol scattering and RH, and transport pathways over the target areas are studied here.



### 4.1 Clean PBL

Fig. 6a and 7a show the vertical profiles of $\sigma_{sca}$ and ambient RH, respectively, retrieved from airborne measurements made under clean PBL conditions during the field campaign. Values of $\sigma_{sca}$ near the surface ranged from 11.7 to 84.5 Mm$^{-1}$ with an average value of 48.5 Mm$^{-1}$. In general, $\sigma_{sca}$ slowly decreased with height, which suggests that the atmosphere was relatively

clean with no distinct aerosol layer identified. The mean $\sigma_{sca}$ profile decreases exponentially with height, which can be expressed as:

$$\sigma_{sca,H}=\sigma_{sca,0}\times exp(-H/H_p), \tag{6}$$

where $\sigma_{sca,0}$ is $\sigma_{sca}$ measured at the surface, H is the altitude, and $H_p$ is the scale height. The ambient RH under clean PBL conditions was divided into two groups: dry (RH: 27.4–36.8%) and humid (RH: 53.1–83.6%). Under dry conditions, $\sigma_{sca}$ and

ambient RH showed a good correlation ($R^2 = 0.62$, Fig. 9a), while under humid conditions, the correlation was low ($R^2 =0.23$). To understand the sources and transport pathways of aerosols over the target areas during the field campaign, we calculated isentropic air mass back trajectories for 72 hours using the NOAA Hybrid Single Particle Lagrangian Integrated Trajectory (HYSPLIT) model (Draxler and Hess, 1997; Stein et al., 2016) at 0.5, 1.5, and 2.5 km above mean sea level. The HYSPLIT model (http://ready.arl.noaa.gov/HYSPLIT.php) was used along with the National Centers for Environmental Prediction's

Global Data Assimilation System 1°×1° meteorological database as input to calculate corresponding air mass backward trajectories terminated at the Xingtai supersite. Fig. 10a showed 72-hour air mass back trajectories under clean PBL conditions during the field campaign. Air masses most commonly originated from the north-western region of the study area. Some of the trajectories can be traced back as far as Mongolia and Siberia, passing over the arid areas to the west of southern Hebei. Some clean and moist air masses originating from the less polluted southern and local areas were also seen.

### 4.2 Pollution in the lower layer of the PBL

The vertical distribution of air pollutants varied greatly from case to case due to a variety of influences. One of the crucial factors was PBL structure, which determines the vertical profile of $\sigma_{sca}$ in the lower troposphere. Fig. 6b shows the vertical profile of $\sigma_{sca}$ under polluted conditions. The corresponding ambient RH profiles (Fig. 7b), $\sigma_{sca}$ as a function of RH (Eq. 7), and the back-trajectory analysis (Fig. 10b) are also shown. The PBL heights of different $\sigma_{sca}$ or ambient RH profiles were

adjusted to the same level to show the similarity of the shapes better. The magnitude of $\sigma_{sca}$ increased slightly with height, suggesting that aerosol pollutants were well mixed with strong turbulence in the PBL. Beyond the PBL, the values of $\sigma_{sca}$ sharply decreased. The PBL heights, as determined by vertical profiles of $\sigma_{sca}$, ranged from 900 to 2000 m with an average value of ~1400 m. The average $\sigma_{sca}$ profile was determined as follows:

$$\sigma_{sca,\,H}=\begin{cases}\sigma_{sca,\,0}\cdot exp\left(-\dfrac{H-H_{PBL}+H_p}{H_p}\right), & (\text{if } H >H_{PBL})\\ \sigma_{sca,\,0}+k\cdot(H-H_{PBL}), & (\text{if } H <H_{PBL})\end{cases}, \tag{7}$$



where $H_{PBL}$ is the altitude of the top of the PBL. Fig. 7b shows the ambient RH profiles under dry and humid conditions. The patterns of dry and humid RH profiles were similar in the PBL, but at the top of the PBL, the RH_dry profile dramatically decreased while the RH_humid profile changed little. Linear fits were made to determine the correlation between RH and $\sigma_{sca}$ (Fig. 9b). When there was pollution in the lower PBL, there was a pronounced correlation between RH profiles (dry and humid) and $\sigma_{sca}$ ($R^2 = 0.78$). The correlation coefficients of the linear regression fit between RH_dry (RH_humid) and $\sigma_{sca}$ above the top of PBL were 0.86 (0.12), which suggests that a linear fit is suitable.

Most back trajectories under polluted conditions originated from the heavily polluted interior and coastal areas south of the study area (Fig. 10b). Some drier air masses (corresponding to RH_dry) were traced back to southern and local areas just as in the case of clean air masses. Moist air masses originated from the clean marine atmosphere to the southeast which then passed over the densely populated eastern/southeastern regions in the free troposphere before reaching the observation site. This analysis of both dry and moist air masses with aerosols in the lowest layer of the PBL shows that heavy local/regional pollution dominated during the field campaign and that the long-range transport of aerosols was not significant.

### 4.3 Pollution in the middle and upper layers of the PBL

In addition to high aerosol concentrations in the lower PBL, upper-layer (referred to as the type A case, an example of which occurred on 2 June 2016) and multi-layer (referred to as the type B case, an example of which occurred on 6 June 2016) aerosol vertical distributions were also observed. In the type A case (Fig. 8a), the vertical profile of aerosol $\sigma_{sca}$ retrieved from airborne measurements shows that the mean $\sigma_{sca}$ near the surface was about 45 Mm$^{-1}$ and varied little below 1 km. A sharp increase in $\sigma_{sca}$ was then observation with a peak value of 200 Mm$^{-1}$ occurring around 2–3 km, above which the $\sigma_{sca}$ profile decreases exponentially with height. For the type B case (Fig. 8b), mean $\sigma_{sca}$ varied greatly with height, which suggests that multiple layers of aerosol particles were present in the PBL. The vertically inhomogeneous distribution of $\sigma_{sca}$ suggests that aerosol particles in the PBL can be significantly affected by both the long-range transport of air pollutants and local emissions in the study area. The profiles of $\sigma_{sca}$ and ambient RH for type A and type B cases have similar shapes with a correlation coefficient of $R^2 = 0.91$ and 0.59, respectively (Fig. 9a and 9b).

As shown in Fig. 10c, back trajectories for the type A case originated from less polluted regions in the northeast China which then moved toward the marine atmosphere over Bohai Bay and the densely populated region east of the study area. The aerosol enrichment in the upper layer of the PBL was probably due to regional transmission and the mixture of anthropogenic aerosols and sea-salt aerosols in the free troposphere. The back trajectories ending at 0.5, 1.5, and 2.5 km over the observation site for the type B case (Fig. 10d) show that air masses originated from the clean marine environment to the southeast, the polluted environment to the southwest, and the eastern coastal region. This could explain the multi-layer vertical distribution of aerosols in the PBL for this case.




## 5 Conclusions

Vertical distributions of aerosol optical properties were characterized using extensive measurements made by airborne and ground-based instruments during an intensive field experiment in May and June of 2016 in the heavily industrialized North China Plain around Xingtai, Heibei Province. During the field campaign, a total of 11 research fights (about 25 flight hours in total) were made as a part of the A$^2$BC and ARIAS experiment. Measurements used in this study include aerosol scattering and backscattering, absorption, extinction, single scattering albedo, Ångström exponent, and AOD. The vertical and regional characteristics of the PBL structure were described to understand their implications on the regional atmospheric environment. Statistical summaries of the vertical distributions of aerosol optical properties focused on four target areas in the NCP region. A total of 38 profiles were compiled and analyzed. Ground-based CAPS and CIMEL retrievals, and meteorological soundings were made at the same time as the airborne measurements to ensure data comparability. Aircraft measurements agree generally well with independent measurements made by radiosonde for temperature and humidity, aerosol extinction from CAPS, and aerosol single scattering albedo from CIMEL sun photometer. While aerosol scattering and extinction coefficients generally decreased with height, there are distinct patterns of the changes with height between clean and polluted episodes. Profiles over the target area showed relatively high values of $\sigma_{sca}$ and $\sigma_{abs}$ suggesting that there were higher concentrations of light-absorbing and scattering pollutants in this region. Mean SSA ($\omega_{\_550}$) ranged from 0.83–0.87 in the four regions, suggesting that moderately strong absorbing aerosols present in the region. The regional mean AOD in each altitude range (< 1 km, 1–2 km, and 2–3 km) was 0.10±0.08, 0.10±0.07, and 0.03±0.03, respectively. Most of the total aerosol concentration in the lower troposphere was found below 2 km during the aircraft campaign.

Three typical PBL structures were identified with distinct airmass transport pathways and the correlation between $\sigma_{sca}$ and RH. In the clean PBL, $\sigma_{sca}$ is strong near the surface and slowly weakens with height. The correlation coefficient of $\sigma_{sca}$ and ambient RH under relatively dry conditions was 0.62. Clean air masses most commonly originated from the northwest, which is far from the study area. When there was pollution in the lower part of the PBL, $\sigma_{sca}$ increased slightly with height then sharply decreased at the top of the PBL. Aerosol scattering and relative humidity showed a good correlation ($R^2 = 0.78$) in the PBL. Most trajectories of this type originated from the heavily polluted interior and coastal areas south and east of the study area. When there was a pollution layer higher into the PBL or multiple layers of pollution, the $\sigma_{sca}$ and ambient RH profiles had similar shapes, the PBL structure could be explained by the source and transmission process of air masses.

*Acknowledgements.* This work was funded by the National Basic Research Program of China "973" (grant no. 2013CB955801), the National Science Foundation of US (grant no. 1558259). We also thank all A$^2$BC and ARIAs research team, especially the flight crew of Hebei Weather Modification Office's Y-12 airplane.



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





**Table 1.** Flight summary.

| Flight number, Date | Time range (UTC) | Profile region | Vertical height, asl (m) |
|---|---|---|---|
| RF1, 20160508 | 02:30–06:32 | JL, QZ, XT | 58–3751 |
| RF2, 20160515 | 04:17–07:04 | JL, QZ | 58–3679 |
| RF3, 20160516 | 07:03–07:54 | | 58–467 |
| RF4, 20160517 | 01:21–03:13 | JL, QZ | 58–2924 |
| RF5, 20160519 | 07:42–09:09 | LC | 58–3733 |
| RF6, 20160521 | 03:57–06:41 | QZ, XT | 58–3242 |
| RF7, 20160528 | 02:16–05:26 | JL, XT | 58–3101 |
| RF8, 20160528 | 08:29–10:24 | XT | 58–3130 |
| RF9, 20160602 | 05:47–06:53 | LC | 58–3591 |
| RF10, 20160606 | 02:09–04:02 | JL | 58–3178 |
| RF11, 20160611 | 02:52–05:45 | LC, XT | 58–3203 |

JL = Julu; LC = Luancheng Airport; QZ = Quzhou; XT = Xingtai

**Table 2.** Instruments used in this study.

| | Instrument | Parameter used in this study | Frequency | Accuracy |
|---|---|---|---|---|
| Airborne platform | Nephelometer, TSI Model 3565, USA | $\sigma_{sca}$, $\sigma_{bsca}$ | 1s | 0.5Mm$^{-1}$ |
| | PSAP, Radiance Research Inc., USA | $\sigma_{abs}$ | 60s | 0.1Mm$^{-1}$ |
| | CWIP, Rain Dynamics, USA | T | 1s | ±1℃ |
| | | RH | 1s | 2% |
| | | Position | 0.1s | |
| Xingtai supersite | CAPS, Aerodyne Res. Inc., USA | $\sigma_{ext}$ | 1s | <2Mm$^{-1}$ |
| | | Pressure | | 0.1hpa |
| | Radiosondes, Graw Model DFM-09, Germany | T | 1s | 0.1℃ |
| | | RH | | 1% |
| | CIMEL, CE-318, France | SSA | Hourly | 0.03 |

**Table 3.** Comparison of AERONET-retrieved and in situ aircraft-measured $\omega_{\_550}$ from this and other studies.

| Location | Period | AERONET (CE-318) | Aircraft in situ (Neph TSI-3536; PSAP) | Reference |
|---|---|---|---|---|
| Northeastern United States | Summer 2011 | 0.979 | 0.99 | (Schafer et al., 2014) |
| South Africa | August to September 2000 | 0.85±0.02 | 0.89±0.03 | (Leahy et al., 2007) |
| West Africa | 19 January 2006 | 0.85 | 0.87 | (Johnson et al., 2010) |
| North China | 28 May 2016 | 0.93 | 0.94 | This work |

**Table 4.** Means and standard deviations of aerosol optical properties over LC, JL, QZ, and XT during the experiment.

| Parameter | LC | JL | QZ | XT |
|---|---|---|---|---|
| $\sigma_{sca}$ (550nm) (Mm$^{-1}$) | 57.1±67.71 | 95.57±106.61 | 87±100.43 | 75.3±84.58 |
| $\sigma_{bsca}$ (550nm) (Mm$^{-1}$) | 7.67±7.6 | 9.91±9.19 | 11.12±10.51 | 9.21±8.6 |



| | | | | |
|---|---|---|---|---|
| $\sigma_{abs}$ (550nm) (Mm$^{-1}$) | 9.06±9.17 | 5.06±4.43 | 6.67±7.34 | 8.13±6.75 |
| $\sigma_{ext}$ (550nm) (Mm$^{-1}$) | 74.24±77.25 | 103.14±110.07 | 100.46±109.24 | 89.56±91.04 |
| $\beta_{sca}$ (550nm) | 0.17±0.21 | 0.15±0.24 | 0.18±0.24 | 0.18±0.31 |
| $\alpha$ (450–700nm) | 1.46±0.97 | 1.57±0.73 | 1.56±0.97 | 1.6±0.81 |
| $\omega$ (550nm) | 0.83±0.11 | 0.87±0.12 | 0.83±0.15 | 0.83±0.15 |

JL = Julu; LC = Luancheng Airport; QZ = Quzhou; XT = Xingtai

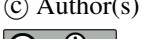


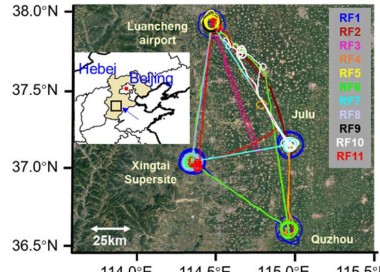

**Fig. 1.** Flight tracks of the 11 research flights conducted over Hebei Province in the North China Plain during the May-June 2016.

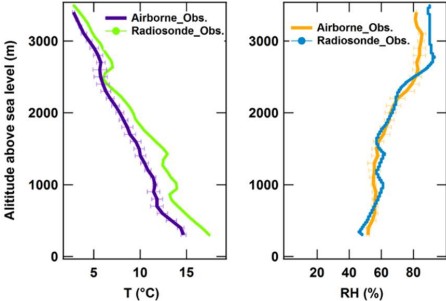

5      **Fig. 2.** Vertical profiles of temperature (T, left) and relative humidity (RH, right) from radiosonde and airborne measurements made on 8
       May 2016 over Xingtai supersite. Horizontal bars represented standard deviations.

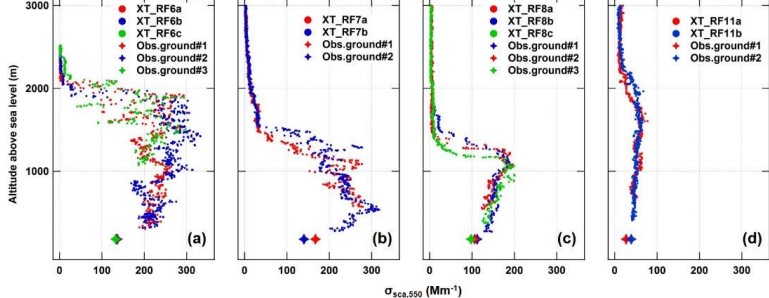

**Fig. 3.** Aircraft-measured vertical profiles (colored circles) and surface CAPS-measured (colored plus symbols) $\sigma_{ext}$ at 550 nm on (a) May
10     21 (RF6, a, b, and c represents spiral up/down), (b) May 28 (RF7), (c) May 28 (RF8), and (d) June 11 (RF11).



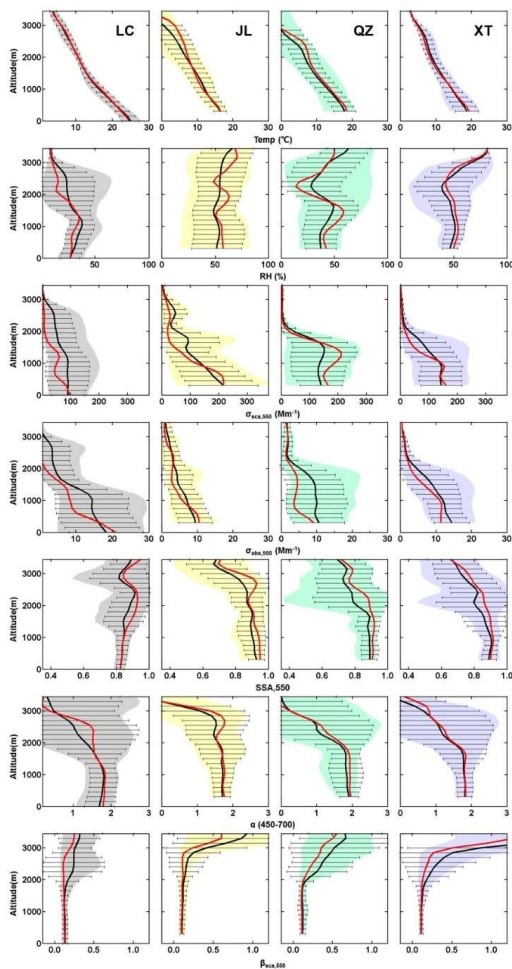

**Fig. 4.** Mean vertical distributions of (from the top row to the bottom row) temperature (°C), relative humidity (%), $\sigma_{sca}$ at 550 nm (Mm$^{-1}$), $\sigma_{abs}$ at 550 nm (Mm$^{-1}$), SSA at 550 nm, Ångström exponent ($\alpha$), and $\beta_{sca}$ at 550nm over LC, JL, QZ and XT (from left column to right column, respectively). Black and red lines represent the mean and the median, respectively, and horizontal bars are standard deviations at every 150-m level. The colored shaded areas represent the 10$^{th}$ and 90$^{th}$ percentiles of the data.





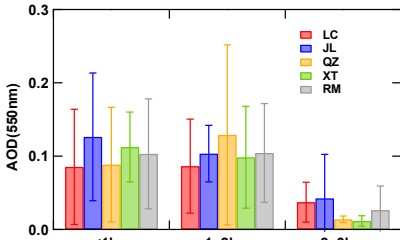

**Fig. 5.** Mean AOD at 550 nm at LC, JL, QZ, and XT, and overall mean AOD (RM) at 550 nm for different altitude ranges (< 1 km, 1–2 km, and 2–3 km). Standard deviations are also shown.

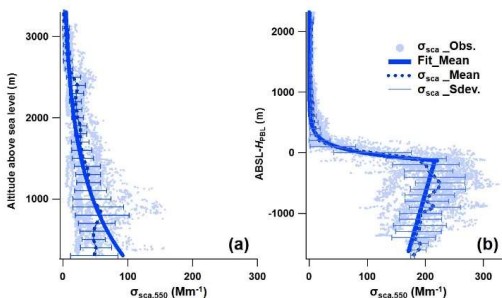

**Fig. 6.** Mean vertical distributions of $\sigma_{sca}$ at 550 nm (in Mm$^{-1}$) during the flight campaign for those cases of (a) clean PBL and (b) pollution in the lower layer of the PBL where PBL heights have been normalized to the same altitude. Blue dashed lines and light blue dots represent mean $\sigma_{sca}$ vertical profiles and 1 s Nephelometer-measured $\sigma_{sca}$, respectively. Thick blue lines show the calculated fitting curves of the $\sigma_{sca}$ profiles (see Eq. 6 and 7) and horizontal error bars represent the standard deviations at every 100 m level.

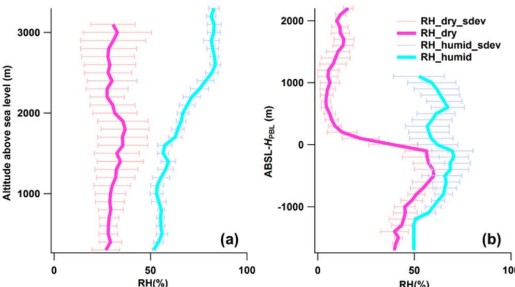

**Fig. 7.** Mean vertical profiles of relative humidity (%) during the flight campaign for those cases of (a) clean PBL and (b) pollution in the lower layer of the PBL where PBL heights have been normalized to the same level. Magenta lines represent data collected under dry





conditions; blue lines represent data collected under humid conditions. Horizontal error bars represent the standard deviations at every 100-m level.

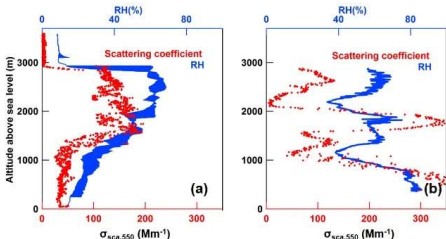

**Fig. 8.** Vertical distributions of $\sigma_{sca}$ at 550 nm (red) and relative humidity (RH, blue) for the enrichment of aerosols in the upper layer of the
5    PBL on 2 June (a) and in multiple layers of the PBL on 6 June (b).

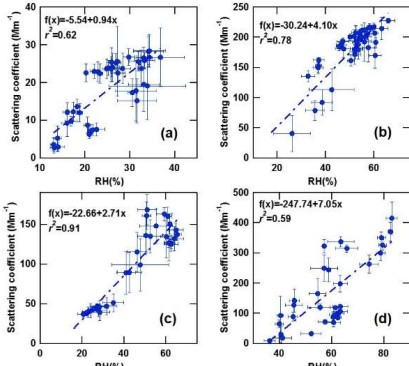

**Fig. 9.** $\sigma_{sca}$ at 500 nm as a function of relative humidity (RH, in %) for those cases when the PBL was (a) clean and dry, (b) polluted in the
lower PBL, (c) polluted in the upper PBL, and (d) polluted in multiple layers of the PBL. The linear regression best-fit lines through the data
10    are shown (dashed lines). The regression relationships and coefficients of determination are given in each panel.



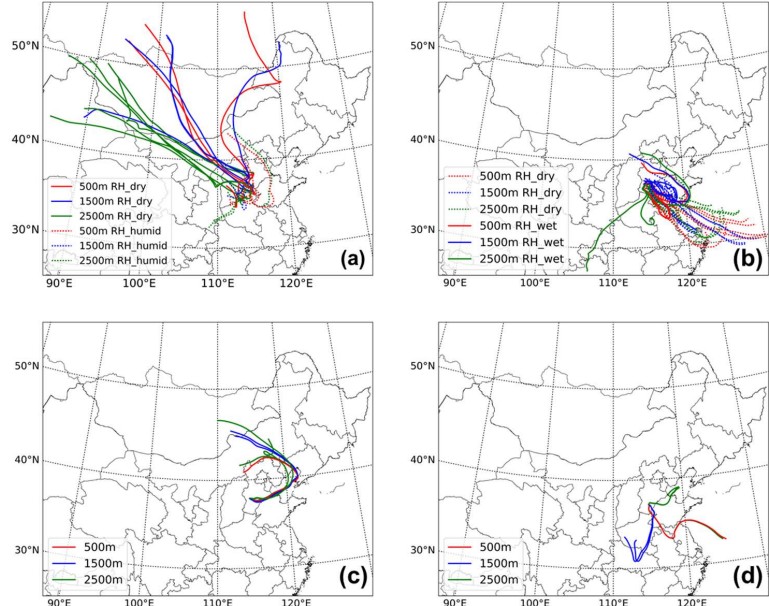

**Fig. 10.** 72 hours HYSPLIT back trajectories for the cases when the PBL was (a) clean, (b) polluted in the lower PBL, (c) polluted in the upper PBL, and (d) polluted in multiple layers of the PBL. Trajectories at different levels under dry (solid lines) and humid (dashed lines) conditions are shown: 500 m (red), 1500 m (blue), and 2500 m (green) above mean sea level.

