# Peer review of "Vertical distributions of aerosol optical properties during the spring 2016 ARIAs airborne campaign in the North China Plain"

_Atmospheric Chemistry and Physics, 2017_

## Referee Comment (RC1) · Anonymous Referee #3 · 22 Jan 2018

General Comments: The paper describes airborne and surface measurements taken around Xingtai, Heibei Province during May-June of 2016 as part of the A2BC and ARIAS field experiments. The manuscript is generally well written and presents important observations over a densely populated area in China that describe aerosol optical properties, and their vertical distributions. For this reason the manuscript should be published. However, the manuscript lacks a clear description of the relevance and motivation for the campaign and its' data. The section (4) of the manuscript that analyzes the relationship between PBL structure and aerosol scattering coefficient is confusing and needs significant revisions.

[Figure]

Specific Comments:

Experimental Description:

Define the geographic limits of the North China Plain.

What is the relevance of the NCP to the rest of China and Northeast Asia (other than being densely populated and fast developing - these two characteristics could be used to describe most anywhere in China)?

What is the motivation behind the location of the surface supersite, the flight paths, locations for spirals, and the frequency of flights?

What is the relevance and motivation for the time period of the flights?

How representative are the measurements of the NCP region during these two months?

What were the general meteorological conditions during the campaign? Clear? Overcast? Stagnant? What are the prevailing winds? Do you expect long-range transport during this time period?

Section 4.1:

How was clean PBL defined? Which flights/spirals/dates were identified as clean?

What is the purpose of Eq 6? Did you calculate the scale height? What is it?

What is the significance of there being high correlation between RH and sigma_scat at low RH, but not at high RH? How many profiles are you basing this correlation off of?

In general, it is not clear what data is plotted in Figure 9.

Section 4.2:

Again, which flights/spirals/dates were identified as polluted? How did you define polluted?

It seems like you fit Eq 7 to the data. Please state this explicitly. What method did you use to derive this fit?

Does the scatter plot in Fig 9b include both dry and humid profiles? Please state explicitly.

You seem to contradict yourself, by first saying that both dry and humid profiles have good correlation between RH and $\sigma$sca, then above the PBL only dry profiles are correlated. Please clarify.

Section 4.3:

Did the upper-layer and multi-layer profiles only occurs on these two days?

At which location were these profiles measured?

Figs 9c and 9d refer to these profiles, not 9a and 9b. Do you only include data from those two dates/profiles in Fig 9c and 9d. Please clarify.

Section 4 in general only considers aerosol scattering. What about absorption? Angstrom exponent? How are these aerosol properties affected by different transport patterns?

Technical Corrections:

Line 17: shows should be show

Line 29: drastically should be drastic

Line 31: 'impact on' should be 'impacts on the'

Line 27: 'scanty' should be 'scant'

Line 27: Should NFS be NSF? Please define.

Line 16: 'includes' should be 'include'

---

## Referee Comment (RC2) · Anonymous Referee #2 · 26 Jan 2018

This study characterized the vertical distributions of aerosol properties using airborne and ground-based measurements over Xingtai, Heibei Province in north China Plain. Given the importance of aerosol vertical structure, this study presents some interesting scientific results, especially the correlations of the vertical aerosol scattering coefficients and relative humidity, based on first-hand observations. The manuscript is overall well-written but lacks of some important introduction about the vertical distributions of aerosols. Also it needs some more description about the experiment and the definitions of a few important parameters. Revisions are needed to address the following questions before the acceptance of this manuscript:

1. Introduction: This study focuses on the vertical distribution of the aerosol properties. However, it lacks of information about the importance of the vertical distribution of aerosols and the uncertainty in the observed aerosol vertical structure. The vertical distribution of aerosols is very important as it modifies the vertical profile of radiative heating in the atmosphere and affects the atmospheric stability and convection. It also influences the radiative effect at the top of the atmosphere (TOA), particularly when the aerosols have strong absorption of solar radiation. A number of field programs have also been carried out to measure the vertical distribution of aerosols.

Please give a literature review about the research that have been conducted in association with aerosol vertical distribution, such as the following work:

Meloni, D., Sarra, A. D., Iotio, T. D., and Fiocco, G.: Influence of the vertical profile of Saharan dust on the visible direct radiative forcing, J. Quant. Spectrosc. Ra., 93, 497–413, 2005.

Gadhavi, H. and Jayaraman, A.: Airborne lidar study of the vertical distribution of aerosols over Hyderabad, an urban site in central India, and its implication for radiative forcing calculations, Ann. Geophys., 24, 2461–2470, doi:10.5194/angeo-242461-2006, 2006.

Johnson, B. T., Heese, B., McFarlane, S. A., Chazette, P., Jones, A., and Bellouin, N.: Vertical distribution and radiative effects of mineral dust and biomass burning aerosol over West Africa during DABEX, J. Geophys. Res., 113, D00C12, doi:10.1029/2008JD009848, 2008.

Zhang, L., Li, Q. B., Gu, Y., Liou, K. N., and Meland, B.: Dust vertical profile impact on global radiative forcing estimation using a coupled chemical-transport–radiative-transfer model, Atmos. Chem. Phys., 13, 7097-7114, https://doi.org/10.5194/acp-13-7097-2013, 2013.

2. Page 3, line 27: What does NFS stand for?

3. Section 2.1, Figure 1: It's hard to tell where the sites are with such a small map. It would be better to give a larger geographic map, at least for China and the coastal area, and then a zoom-in map for north China Plain and the sites.

4. Section 2: Is this the first time that the ARIAs project is introduced (I didn't see any reference). If yes, I would suggest that a little more information should be given to describe the scientific objective of this project and justify how the super site in Xingtai was chosen.

5. Section 4: What is the definition for clean or polluted PBL, e.g., using a critical value of AOD within PBL? What is the scale height Hp in this study? Normally it represents the height when the aerosol is reduced to 1/e of its surface value. Is it a prescribed value or determined from the observation? And how is PBL height determined from scattering coefficient in this study?

6. Section 4.1 & 4.2: What ambient RH is used to determine the cases as dry/humid conditions, e.g., the average RH within PBL or the RH at a certain level? What is the percentage of the dry and humid cases?

Figs. 6b & 7b are interesting. Since Fig. 7b is done by separating dry and humid conditions, it would be interesting to see Fig. 6b in dry and humid conditions as well.

[Figure]

---

## Referee Comment (RC3) · Anonymous Referee #1 · 8 Feb 2018

Overall comment: This manuscript presents a study of aerosol optical and vertical distribution over the NCP region using aircraft measurements and ground based data obtained from 2016 ARIAs airborne campaign. Among many recent research on the Chinese air pollution, this manuscript provides a new insight on the linkage between PBL structure and aerosol optical properties in vertical prospective. Although the instruments and methodology is not new, the integrational efforts on a suite of instruments equipped to Y-12 aircraft and cooperating with surface measurements should be recognized its importance. The group have rich experiment (many papers published) on aerosol observations as well as airborne measurements, turns out some descriptions on the methodology have been simplified, and scientific discussions tend to draw con-

clusions quickly without strong statement. For example, there is lots of room for the Table 3 SSA comparison discussion, especially the main goal of this paper is to discuss the consistency of airborne and surface-based measurements (Page 3, line 8). The paper is generally well-written and the reviewer does not have difficulty to follow the context. Following comments are provided where I think the paper needs improvement, especially at some places more clarity and insights are needed. My summary recommendation to the Editor is that this paper be major revision, and that the paper be recirculated for secondary review after revision.

General Comments: (1) The aerosol inlet system on the aircraft should be discussed in the manuscript. It strong related to how to evaluate the accuracy of airborne aerosol measurement. (2) SSA is a very sensitive and unstable parameter. Authors should provide the detail descriptions on how to calculate the columnar SSA from airborne measurement data and their uncertainties? When different aerosol types or multi aerosol layer in vertical, how authors apply to the calculation? (3) Expending the discussion on the Table 3. Why only 28 May, 2016 case shown in the table? Maybe the scatter plot of AERONET SSA vs Aircraft in situ for each flights can provide more strong conclusion. (4) How to calculate AOD for different layers (Fig 5) should describe in the manuscript. (5) In figure 5(b), it is not clear how authors define the PBL height? Although in the manuscript, the authors say they determined by aerosol scatter profile (Page 9, line 28), does that quantitatively define PBL height? In figure 5(b), how does authors apply the normalization? Does use 1400m PBL height or individual PBL height from each profiles? (6) It is not clear why the authors use the scale height (eq 6 and 7) here? Is that only to calculate the mean profile (or Fit mean in Fig 6)? (7) RH and scattering coefficient have strong correlation. Does mean that the PBL aerosol is well mixed and the different in aerosol scatter is due to aerosol hygroscopcity?

Specific Comments: (1) Figure 2a: why the large discrepancy between airborne and sounding data? In the airborne T profile, one can not easy determine the inversion layer. (2) Equation (1): what is the r value? (3) Equation (2): what is the C value? B

is beta right? (4) Does adjusted Beta sca (Beta sca_adj) use in the follow analysis? If yes, author should maintain the symbol consistency. (5) Page 5 line 4: for loading? (not clear) (6) The equation should better insert in a paragraph, not like equations (4) and (5). (7) Equation (6): what is Hp value used? (8) Equation (7): What is k value? (9) Page 10 (lines 5-6): The correlation.........is suitable, please remove since not necessary (not relevant) to discuss here. (10) Page 10, line 21: Why affected by the long-range transport? (11) Page 10, line 23: Figure 9c, 9d.

---

## Author Comment (AC2) · 13 Apr 2018

**Anonymous Referee #1:**

We sincerely appreciate for your time and attention on our paper. The comments and suggestions you gave are very helpful for us to improve our paper. We now present point-by-point replies (in black) to all your comments (in green) in this response and the corresponding changes in the revised manuscript have been highlighted in blue.

● **Experimental Description:**

*1. Define the geographic limits of the North China Plain.*

**Reply:** The NCP region includes Beijing, Tianjin, and most of Hebei province, which is surrounded by Taihang Mountains to the west, Yanshan Mountains to the north, and Bohai Sea and Yellow Sea to the east with Korea and Japan farther east of the Yellow Sea.

We have added a China Map to show the location of the North China Plain region in the revised manuscript.

[Figure]

**Fig.1 Map of the geographic location of North China Plain and the Xingtai supersite (a), and the flight tracks of the 11 research flights conducted over Hebei Province in May-June 2016 (b).**

*2. What is the relevance of the NCP to the rest of China and Northeast Asia (other than being densely populated and fast developing - these two characteristics could be used to describe most anywhere in China)?*

**Reply:** The NCP region is the most polluted area in China and Northeast Asia. With great industrial activity and increase in automobiles during the last several decades, the consumption of fossil fuels like coal, gasoline, diesel, and natural gas has increased dramatically in this region. The combustion of fossil fuels emits large amounts of particulate and gaseous pollutants into the atmosphere, leading to substantial environmental problems. Industrial and automobile related primary emissions, as well as the formation of secondary aerosols, combined with the transport of dust from the desert region in Northwest China, result frequently in heavy aerosol loads in the NCP depending on the meteorological conditions. In this area, air quality is significantly influenced by aerosol particles and high loads of aerosol

pollution severely degrade visibility, especially under high RH conditions.

Besides being one of the most populated region with fast economic development that generate huge anthropogenic emissions, the region has a unique topography which, together with weather regimes, plays an important role in regulating air pollution and in return influences the regional climate. With a mountain range to the west (Yan and Taihang mountains), any easterly and southerly winds may bring in and accumulate pollutants in both particulate and gas phases. Blocking of the polluted air mass may not only affect the boundary-layer condition by altering thermodynamic state but also induce a conditional instability to induce heavy rainfall as reported in another study of similar topographic setting (Fan et al., 2015). These features make the NCP to be a unique area to study emissions of air pollutants and their transport to surrounding areas in China and Northeast Asia.

**3. *What is the motivation behind the location of the surface supersite, the flight paths, locations for spirals, and the frequency of flights?**

**Reply:** There are a few reasons we choose Xingtai as a surface supersite station. First of all, it represents the typical geographical features and air pollution characteristics of the NCP regions, as shown in the responses to Comments 1& 2s. Statistics shows that Xingtai and other southern Hebei cities were among the worst-ranked cities by seasonal or annual air quality index in 2013 and 2014 (Li et al., 2015). Secondly, as a national primary weather station, basic meteorological parameters and sounding observations have been made at the Xingtai supersite routinely. Lastly, it has existing infrastructure to support the field experiment.

As the base airport for this experiment was located in Luancheng, Shijiazhuang, and considering the combination of aircraft and surface observations, most of our research flights, focus on vertical distribution of air pollutants, were conducted over Luancheng and Xingtai. Due to restrict control of airspace in this area, we choose Julu and Quzhou as different underlying surface conditions for comparison. As we usually conduct observation of aerosol optical properties under cloudless day, the flight design and its frequency were determined by weather conditions and airspace availability.

**4. *What is the relevance and motivation for the time period of the flights?**

**Reply:** The overall period of our flights is dictated by the duration of the entire experiment that was chosen in light both the regional climate and in cooperation with another major international experiment, the NASA KORUS-DAQ campaign, in downwind South Korea (Al-Saadi et al., 2016). The dates and time of any individual flights were determined by weather regimes and aerospace availability, and often the latter being more crucial due to difficulties in getting any. Weather-wise, NCP region is characterized by a warm temperate monsoon climate, dominated by the northern cyclones and cold fronts in the spring, resulting in strong wind, and precipitation by convection is mainly concentrated in summer (Sun et al., 2013). To avoid the strong wind and convective precipitation, the experiment was carried out in May and June,

which could essentially represent the aerosol conditions from spring to summer in the NCP region.

**5. *How representative are the measurements of the NCP region during these two months?**

**Reply:** Two experimental aircrafts were deployed during the in-situ campaign, a Y-12 and a Cheyenne-III (C-III). The Y-12 flights mainly focused on atmospheric pollutants under cloud-free conditions. The Cheyenne-III was equipped with a suite of instruments for cloud and precipitation measurements. The following table lists weather conditions during the experiment. We carried out the flight experiments under a variety of weather (cloudy, rainy, or strong wind) and air pollution (clear, hazy, heavily polluted) conditions over the 2-month period, so the measurements are representative for the level of pollution and aerosol optical properties in the NCP region.

**Table 1 An overview of flight information and the weather phenomenon by ground measurement during the experimental period.**

| date | flight or not | | weather of LC Airport | date | flight or not | | weather of LC Airport |
|------|------|-------|-----------------------|------|------|-------|-----------------------|
|      | Y-12 | C-III |                       |      | Y-12 | C-III |                       |
| 0508 | √    |       | haze                  | 0526 |      |       | clear                 |
| 0509 |      |       | light rain            | 0527 |      |       | overcast              |
| 0510 |      |       | haze                  | 0528 | √√   | √     | clear                 |
| 0511 |      |       | heavy polluted        | 0529 |      |       | clear                 |
| 0512 |      |       | overcast              | 0530 |      |       | haze                  |
| 0513 |      |       | overcast              | 0531 |      |       | clear                 |
| 0514 |      |       | rainfall              | 0601 |      |       | haze                  |
| 0515 | √    |       | clear                 | 0602 | √    |       | heavily polluted      |
| 0516 | √    |       | clear                 | 0603 |      | √     | overcast              |
| 0517 | √    |       | haze                  | 0604 |      | √     | rainfall              |
| 0518 |      |       | haze                  | 0605 |      | √√    | rainfall              |
| 0519 | √    |       | heavily polluted      | 0606 | √    |       | haze                  |
| 0520 |      |       | haze                  | 0607 |      |       | heavily polluted      |
| 0521 | √    | √     | haze and light rain   | 0608 |      |       | haze                  |
| 0522 |      |       | haze                  | 0609 |      |       | heavily polluted      |
| 0523 |      | √     | rainfall              | 0610 |      |       | thunder               |
| 0524 |      |       | very strong wind      | 0611 | √    |       | clear                 |
| 0525 |      |       | overcast              |      |      |       |                       |

**6. *What were the general meteorological conditions during the campaign? Clear? Overcast? Stagnant? What are the prevailing winds? Do you expect long-range transport during this time period?**

**Reply:** The meteorological conditions during the flight campaign are shown in Table 1. Generally, we carried out the aerosol optical observations during daytime under variable air quality conditions from very clean to rather hazy conditions. The

prevailing winds were northwesterly or westly that brought dry and clean airmasses from less polluted areas to the experimental region. The long-range transport of air pollutants occurred when the wind direction changed. For example, easterly or southerly winds brought pollutant plumes to the study region, which lead to the occurrence of pollution events. Similarly, under weak wind conditions, the stagnant conditions caused multi-day air pollution episode.

● **Section 4.1:**

*7. How was clean PBL defined? Which flights/spirals/dates were identified as clean?*

**Reply:** We defined the clean PBL where the mean value of $\sigma_{sca}$ below 100 Mm$^{-1}$ at every 100m and low levels of gas pollutants like CO and NOx were observed. Based on this definition, the clean flights/spirals on different dates include RF1 (on May 8, spirals at JL, XT), RF4 (on May 17, spirals at JL, QZ) and RF11 (on June 11, spirals at LC, XT), about 13 vertical profiles in total.

*8. What is the purpose of Eq 6? Did you calculate the scale height? What is it?*

**Reply:** The purpose of Eq.6 was considered as follows. When the $\sigma_{sca}$ profiles in clean PBL cases are taken as an ensemble, and they fit nicely on a diagram, the $\sigma_{sca}$ profiles are transformed into a parameterized model. The scale height is now calculated in Eq. (8) as the height when $\sigma_{sca}$ is reduced to 1/e of its surface value. We have added the following description in Section 4.1 of the revised manuscript:

The mean $\sigma_{sca}$ profile decreases exponentially with height, which can be expressed as:

$$\sigma_{sca,H}=\begin{cases}\sigma_{sca,0}\cdot\exp(-(H-H_{RS})/H_p), & (\text{if } H > H_{RS})\\ \sigma_{sca,0}, & (\text{if } H \leq H_{RS})\end{cases}, \qquad (6)$$

where $\sigma_{sca,0}$ is $\sigma_{sca}$ measured at the surface, H is the altitude above sea level, and $H_p$ is the aerosol scale height ($H_p$ represents the height when $\sigma_{sca}$ is reduced to 1/e of its surface value); $H_{RS}$ represent a relative stable layer near surface where the vertical variation of $\sigma_{sca}$ was not significant. In the cases of a typical clean PBL, $\sigma_{sca,0}$=124 Mm$^{-1}$, $H_p$ =1146 m and $H_{RS}$ =837 m. The linear regression analysis suggests a correlation coefficient, $r^2$=0.96.

*9. What is the significance of there being high correlation between RH and sigma_scat at low RH, but not at high RH? How many profiles are you basing this correlation off of?*

**Reply:** The establishment of the correlation between $\sigma_{sca}$ and RH, mainly considering the vertical distribution of aerosol optical properties (hard to get) could be calculated by means of the measurement of RH profiles (easy available) during this field campaign. We can also estimate the source of corresponding air masses by the back-trajectory analysis.

In the clean PBL conditions, there were 11 profiles in total basing this correlation. We can infer that if the measured RH profile was in accordance with RH_dry profile in

Fig.6c (in the revised manuscript), the $\sigma_{sca}$ distribution was shown in correspond in Fig.6a, and the air mass most commonly originated from Northwest China by long-range transport. There was a poor correlation ($r^2$=0.23) between RH and $\sigma_{sca}$ in the case of higher RH condition (RH>50), but we could conclude that the surface layers are relatively stable and the pollutants were mainly from emissions in local or the surrounding areas.

**10.** *In general, it is not clear what data is plotted in Figure 9.*

**Reply:** We make a linear regression analysis between airborne measured ambient RH and $\sigma_{sca}$ (550nm) profiles in Fig. 9 (now Fig. 8 in the revised manuscript). The scatter, the horizontal and vertical error bars represent mean values and the standard deviations of RH and $\sigma_{sca}$ at every 100m level. For example, under clean PBL condition, the correlation between RH_dry profile (pink line in Fig.6c) and its corresponding $\sigma_{sca}$ was $r^2$=0.62, where r is the correlation coefficient (Fig.8a).

- **Section 4.2:**

**11.** *Again, which flights/spirals/dates were identified as polluted? How did you define polluted?*

**Reply:** Polluted flights/spirals on different days include RF5 (on May 19, spirals at LC), RF6 (on May 21, spirals at QZ, XT), RF7 (on May 28, spirals at JL, XT), RF8 (on May 28, spirals at XT), and 21 vertical profiles in total were identified as pollution in the lower layer of PBL. We define polluted PBL by the mean value of $\sigma_{sca}$ greater than 100 $Mm^{-1}$ in the PBL layer and high levels of gas pollutants like CO and NOx were observed. Weather phenomenon (Table 1) at flight time was also one of the considered factor. Three typical types of polluted PBL are classified according to the shapes of $\sigma_{sca}$ vertical profiles. Fig.6b shows the vertical profiles of $\sigma_{sca}$ of pollution in the lower layer of the PBL. The profiles of this type show that the gradient of $\sigma_{sca}$ are generally small from surface to a certain layer ($H_{PBL}$), then the value of $\sigma_{sca}$ sharply decreased.

**12.** *It seems like you fit Eq 7 to the data. Please state this explicitly. What method did you use to derive this fit?*

**Reply:** As mentioned in previous paragraphs, the structure of PBL is a crucial factor, which determines the vertical distribution of aerosol property in the lower troposphere. The types of PBL structure are classified by the shapes of $\sigma_{sca}$ profiles. To better show the similarity of these profile shapes, the PBL heights of different profiles have been adjusted to the same level (Fig.6b).

According to previous study of aerosol vertical distribution, e.g. Liu et al. (2009), we use linear fit to represent the $\sigma_{sca}$ distribution in the PBL and use curve fitting (exponential function) to represent $\sigma_{sca}$ profiles in the free troposphere.

We have added the following description in Section 4.2 in the revised manuscript.

$$\sigma_{sca,\,H}=\begin{cases}\sigma_{sca,\,PBL}\cdot exp(-(H-H_{PBL})/\,H_p),&(if\,H>H_{PBL})\\\sigma_{sca,\,0}+k\cdot H,&(if\,H\le H_{PBL})\end{cases},\tag{9}$$

where $H_{PBL}$ is the normalized altitude of PBL height, $H_p$ is the aerosol scale height in the free troposphere, k is the changing rate of $\sigma_{sca}$ in the PBL. In these cases, $\sigma_{sca,0}$=171 $Mm^{-1}$, $H_p$ =216 m, k= 0.03 $Mm^{-1}$ $m^{-1}$ and $r^2$=0.9394. Fig. 6d shows the ambient RH profiles under dry and humid conditions. The shapes of dry and humid RH profiles were similar in the PBL, but at the top of the PBL, the RH_dry profiles dramatically decreased while the RH_humid profiles slightly changed. Linear fits were made to determine the correlation between RH and $\sigma_{sca}$. Under dry condition, there was a pronounced correlation ($r^2$=0.95) between RH_dry and $\sigma_{sca}$ profiles. But under humid condition, the correlation coefficient was 0.12, which suggest a poor correlation between RH_humid and $\sigma_{sca}$ profiles.

*13. Does the scatter plot in Fig 9b include both dry and humid profiles? Please state explicitly.*

**Reply:** Yes, the scatter plot in Fig.9b (now Fig.8b in the revised manuscript) include both dry and humid profiles, but only in the lower layer of PBL (H < $H_{PBL}$). We have made a change to Fig. 8 and added some description in Section 4.2 in the revised manuscript. Now, the Fig.8b showed a linear regression analysis and the correlation coefficient between RH_dry and $\sigma_{sca}$ profiles (corresponding to the blue line of RH_dry in Fig. 6d).

*14. You seem to contradict yourself, by first saying that both dry and humid profiles have good correlation between RH and σsca, then above the PBL only dry profiles are correlated. Please clarify.*

**Reply:** Thanks for your comment. As we mentioned above, under dry condition, there was a pronounced correlation ($r^2$=0.95) between RH_dry and $\sigma_{sca}$ profiles. While under humid condition, the correlation coefficient was 0.12, which suggest a poor correlation between RH_humid and $\sigma_{sca}$ profiles.

However, if the constraint is PBL height,the correlation between RH and $\sigma_{sca}$ was $r^2$=0.78 (below PBL height, H≤$H_{PBL}$) and $r^2$=0.07 (above PBL height, H>$H_{PBL}$), both of them including dry and humid profiles. To better explain it, we have added the following description in Section 4.2 in the revised manuscript.

"Linear fits were made to determine the correlation between RH and $\sigma_{sca}$. Under dry condition, there was a pronounced correlation ($r^2$=0.95) between RH_dry and $\sigma_{sca}$ profiles. But under humid condition, the correlation coefficient was 0.12, which suggest a poor correlation between RH_humid and $\sigma_{sca}$ profiles."

- **Section 4.3:**

*15. Did the upper-layer and multi-layer profiles only occurs on these two days?*

**Reply:** Yes, we found such types of PBL structure only on these two days during the experimental period and each type includes 2 profiles.

*16. At which location were these profiles measured?*

**Reply:** The profiles were measured at LC (RF9 on June 2 with 2 spirals) and JL (RF10 on June 6 with 2 spirals), respectively. We have included the location information in caption of Fig. 7.

*17. Figs 9c and 9d refer to these profiles, not 9a and 9b. Do you only include data from those two dates/profiles in Fig 9c and 9d. Please clarify.*

**Reply:** Thanks for your comment, we correct the mistake in the revised manuscript. The Fig.9c and 9d (now Fig. 8c and 8d) include data measured at LC (RF9, 2 profiles) and JL (RF10, 2 profiles). We have clarified the two dates in the caption of Fig. 8. The scatter, the horizontal and vertical error bars represent mean values and the standard deviations of RH and $\sigma_{sca}$ at every 100 m level.

*18. Section 4 in general only considers aerosol scattering. What about absorption? Angstrom exponent? How are these aerosol properties affected by different transport patterns?*

**Reply:** The $\sigma_{abs}$ and Angstrom exponent profiles are shown in the following figure. According to the polluted conditions mentioned above, four types of profiles are classified by their different shapes. Liner regression shows the correlation between $\sigma_{abs}$ and its corresponding ambient RH are $r^2$=0.24, 0.62, 0.57, 0.31, respectively, which was not exactly the same as $\sigma_{sca}$ we discussed in the paper. Furthermore, the $\sigma_{abs}$ scatters in second type under different RH condition, $r^2$=0.66 (between $\sigma_{abs}$ and RH, under dry condition) and $r^2$=0.01 (under humid condition). Compared with $\sigma_{sca}$, the $\sigma_{abs}$ has less correlation with ambient RH. Similar method was used to discuss the Angstrom exponent profiles in Fig.2, but the correlation is substantially poorer from the result.

[Figure]

**Fig.2 Different types of aircraft-measured vertical profiles of $\sigma_{abs}$ (green line) and Angstrom exponent (purple line). Horizontal error bars represent the standard deviations at every 100m level.**

Cluster analysis of 72 hours HYSPLIT back-trajectories were carried out to discuss the influence of aerosol optical properties by different transport pattern. Taking absorption coefficient as an example, the long-range transport from north (89%) was the dominate type and the local air mass contribute a fraction of $\sigma_{abs}$ (8%), as shown in Fig.3a (corresponding to type I in Fig.2). In contrast, if $\sigma_{abs}$ profile was approximate to the profiles in Fig.2 (II), the enrichment of light-absorbing aerosols in the upper layer of the PBL show that the moist and polluted air masses from interior and coastal areas are dominated (83%) during the field campaign. However, the long-range transport of aerosols was not significant.

[Figure]

**Fig.3 Cluster analysis of 72 hours HYSPLIT back trajectories for the $\sigma_{abs}$ profiles in Fig.2 (a:**

**corresponding to I type and b: corresponding to II type)**

- **Technical Corrections:**

**Reply:** Thanks for your careful review. These have been corrected in the revised manuscript.

**References:**

Fan, J., Rosenfeld, D., Yang, Y., Zhao, C., Leung, L. R., and Li, Z.: Substantial contribution of anthropogenic air pollution to catastrophic floods in Southwest China, Geophysical Research Letters, 42, 6066-6075, 2015.

Li, X., Zhang, Q., Zhang, Y., Zheng, B., Wang, K., Chen, Y., Wallington, T. J., Han, W., Shen, W., and Zhang, X.: Source contributions of urban PM 2.5 in the Beijing–Tianjin–Hebei region: Changes between 2006 and 2013 and relative impacts of emissions and meteorology, Atmospheric Environment, 123, 229-239, 2015.

Liu, P., Zhao, C., Zhang, Q., Deng, Z., Huang, M., Xincheng, M. A., and Tie, X.: Aircraft study of aerosol vertical distributions over Beijing and their optical properties, Tellus Series B-chemical & Physical Meteorology, 61, 756–767, 2009.

Sun, X., Yin, Y., Sun, Y., Sun, Y., Liu, W., and Han, Y.: Seasonal and vertical variations in aerosol distribution over Shijiazhuang, China, Atmospheric Environment, 81, 245-252, 2013.

---

## Author Comment (AC3) · 13 Apr 2018

**Anonymous Referee #2:**

We sincerely appreciate for your time and attention on our paper. The comments and suggestions you gave are very helpful for us to improve our paper. We now present point-by-point replies (in black) to all your comments (in green) in this response and the corresponding changes in the revised manuscript have been highlighted in blue.

1. *Introduction: This study focuses on the vertical distribution of the aerosol properties. However, it lacks of information about the importance of the vertical distribution of aerosols and the uncertainty in the observed aerosol vertical structure. The vertical distribution of aerosols is very important as it modifies the vertical profile of radiative heating in the atmosphere and affects the atmospheric stability and convection. It also influences the radiative effect at the top of the atmosphere (TOA), particularly when the aerosols have strong absorption of solar radiation. A number of field programs have also been carried out to measure the vertical distribution of aerosols. Please give a literature review about the research that have been conducted in association with aerosol vertical distribution, such as the following work:*

**Reply:** Thanks for your comment. We further optimize and make supplement for Section 1 in the revised manuscript.

"Such variable aerosol vertical distributions can alter the optical properties of aerosols such as AOD, thus affecting the regional radiation balance (Liu et al., 2012) and even the global radiative forcing estimation (Zhang et al., 2013). A number of field programs were carried out to measure the vertical distribution of dust or biomass burning aerosols by airborne and surface-based instruments(Johnson et al., 2008). Combine with a radiative transfer model, the radiative effects including aerosol optical properties (Gadhavi and Jayaraman, 2006) and absorption of solar radiation at the top of atmosphere (TOA) (Meloni et al., 2005)could be calculated accurately."

2. *Page 3, line 27: What does NFS stand for?*

**Reply:** "NFS" should be "NSF", which stands for National Science Foundation. We have corrected it in the revised manuscript.

3. *Section 2.1, Figure 1: It's hard to tell where the sites are with such a small map. It would be better to give a larger geographic map, at least for China and the coastal area, and then a zoom-in map for north China Plain and the sites.*

**Reply:** Thanks for your comment, we add a China Map (Fig. 1) to show the location of North China Plain, the coastal area and the Xingtai Supersite in the revised manuscript.

[Figure]

**Fig.1 Map of the geographic location of North China Plain and the Xingtai supersite (a), and the flight tracks of the 11 research flights conducted over Hebei Province in May-June 2016 (b).**

*4. Section 2: Is this the first time that the ARIAs project is introduced (I didn't see any reference). If yes, I would suggest that a little more information should be given to describe the scientific objective of this project and justify how the super site in Xingtai was chosen.*

**Reply:** To better explain the background of this study and provide information about the jointed campaign, we added the following paragraph to the Introduction section.

"The overall goal of the ARIAs project is to integrate *in-situ* observations, satellite remote sensing, and chemical transport models to characterize and quantify tropospheric chemistry and composition over the NCP and to improve modeling tools that can be utilized routinely to eventually evaluate the effectiveness of air pollutant reduction policy scenarios. The trace gases and aerosols have major consequences for downwind areas such as Japan and South Korea, and even for North America."

We choose Xingtai as a surface supersite station for following reasons. First of all, it represents the typical geographical features and air pollution characteristics of the NCP regions, the most polluted area in China and Northeast Asia. Statistics shows that Xingtai and other southern Hebei cities were among the worst-ranked cities by seasonal or annual air quality index during 2013 and 2014 (Li et al., 2015). Secondly, as a national primary weather station, basic meteorological and sounding observations have been made at the Xingtai supersite. Lastly, it has existing infrastructure to support the field experiment.

*5. Section 4: What is the definition for clean or polluted PBL, e.g., using a critical value of AOD within PBL? What is the scale height Hp in this study? Normally it represents the height when the aerosol is reduced to 1/e of its surface value. Is it a prescribed value or determined from the observation? And how is PBL height determined from scattering coefficient in this study?*

**Reply:** As the main focus of this section is aerosol scattering, we defined the clean or polluted PBL by $\sigma_{sca}$ values and the shape of $\sigma_{sca}$ vertical profiles. Weather phenomenon on flight days is another primary factor in our consideration. As clean PBL for example, the mean value of $\sigma_{sca}$ at every 100 m should less than 100 Mm$^{-1}$

(except the surface layer) and decreases exponentially with altitude. Fig.6b shows the vertical profiles of $\sigma_{sca}$ in the lower portion of the PBL. The profiles of this type show that the gradient of $\sigma_{sca}$ are generally small from surface to a certain layer ($H_{PBL}$), then the value of $\sigma_{sca}$ sharply decreased.

We use scale height ($H_p$) as one of parameters to describe a parameterized model of $\sigma_{sca}$ distribution. $H_p$ is determined from airborne observations.

The PBL height is determined by the shapes of $\sigma_{sca}$ vertical profiles. When the pollution in the lower layer of the troposphere, the magnitude of $\sigma_{sca}$ increased slightly with height until a layer where $\sigma_{sca}$ sharply decreased. In this study, the mean decreasing rate is about 0.81 $Mm^{-1} m^{-1}$, We defined the bottom of this layer as the PBL height ($H_{PBL}$).

We have added the following description in Section 4.2 in the revised manuscript.

$$\sigma_{sca, H}=\begin{cases} \sigma_{sca, PBL} \cdot \exp(-(H-H_{PBL})/ H_p), & (if\ H > H_{PBL}) \\ \sigma_{sca, 0} + k\cdot H, & (if\ H \leq H_{PBL}) \end{cases} , \qquad (9)$$

where $H_{PBL}$ is the normalized altitude of PBL height, $H_p$ is the aerosol scale height in the free troposphere, k is the increasing rate of $\sigma_{sca}$ in the PBL. In these cases, $\sigma_{sca,0}$=171 $Mm^{-1}$, $H_p$ =216m, k= 0.03$Mm^{-1}\ m^{-1}$ and $r^2$=0.9394. Fig. 6d shows the ambient RH profiles under dry and humid conditions. The shapes of dry and humid RH profiles were similar in the PBL, but at the top of the PBL, the RH_dry profiles dramatically decreased while the RH_humid profiles slightly changed. Linear fits were made to determine the correlation between RH and $\sigma_{sca}$. Under dry condition, there was a pronounced correlation ($r^2$=0.95) between RH_dry and $\sigma_{sca}$ profiles. But under humid condition, the correlation coefficient was 0.12, which suggest a poor correlation between RH_humid and $\sigma_{sca}$ profiles.

*6. Section 4.1 & 4.2: What ambient RH is used to determine the cases as dry/humid conditions, e.g., the average RH within PBL or the RH at a certain level? What is the percentage of the dry and humid cases? Figs. 6b & 7b are interesting. Since Fig. 7b is done by separating dry and humid conditions, it would be interesting to see Fig. 6b in dry and humid conditions as well.*

**Reply:** We determined the dry/humid case by the mean values of RH vertically. In Section 4.1, the percentage of the dry and humid cases are 0.53 and 0.47, respectively; in Section 4.2, the percentage of the dry and humid cases are 0.67 and 0.33 respectively

We make a change for Fig. 6b by separating dry and humid conditions in the revised manuscript as the following:

[Figure]

**Fig. 6.** Mean vertical distributions of $\sigma_{sca}$ at 550 nm (in Mm$^{-1}$) and relative humidity (%) during the flight campaign for those cases of (a, c) clean PBL and (b, d) pollution in the lower layer of the PBL where PBL heights have been normalized to the same altitude. Grey dashed lines represent mean $\sigma_{sca}$ vertical profiles, the light pink and blue dots represent 1 s Nephelometer-measured $\sigma_{sca}$, under dry of humid condition respectively. Thick lines show the calculated fitting curves of the $\sigma_{sca}$ profiles (see Eq. 6 and 7). Magenta and blue lines represent RH data collected under dry or humid conditions (c, d). The horizontal error bars represent the standard deviations at every 100 m level.

**References:**

Gadhavi, H., and Jayaraman, A.: Airborne lidar study of the vertical distribution of aerosols over Hyderabad, an urban site in central India, and its implication for radiative forcing calculations, Annales Geophysicae, 2006, 2461-2470,

Johnson, B., Heese, B., McFarlane, S. A., Chazette, P., Jones, A., and Bellouin, N.: Vertical distribution and radiative effects of mineral dust and biomass burning aerosol over West Africa during DABEX, Journal of Geophysical Research: Atmospheres, 113, 2008.

Li, X., Zhang, Q., Zhang, Y., Zheng, B., Wang, K., Chen, Y., Wallington, T. J., Han, W., Shen, W., and Zhang, X.: Source contributions of urban PM 2.5 in the Beijing–Tianjin–Hebei region: Changes between 2006 and 2013 and relative impacts of emissions and meteorology, Atmospheric Environment, 123, 229-239, 2015.

Liu, J., Zheng, Y., Li, Z., Connor, F., and Maureen, C.: Seasonal variations of aerosol optical properties, vertical distribution and associated radiative effects in the Yangtze Delta region of China, Medicină Internă, 13, 933-945, 2012.

Meloni, D., Di Sarra, A., Di Iorio, T., and Fiocco, G.: Influence of the vertical profile of Saharan dust on the visible direct radiative forcing, Journal of Quantitative Spectroscopy and Radiative Transfer, 93, 397-413, 2005.

Zhang, L., Li, Q., Gu, Y., Liou, K., and Meland, B.: Dust vertical profile impact on global radiative forcing estimation using a coupled chemical-transport-radiative-transfer model, Atmospheric Chemistry & Physics, 13, 2013.

---

## Author Comment (AC4) · 13 Apr 2018

**Anonymous Referee #1:**

We sincerely appreciate for your time and attention on our paper. The comments and suggestions you gave are very helpful for us to improve our paper. We now present point-by-point replies (in black) to all your comments (in green) in this response and the corresponding changes in the revised manuscript have been highlighted in blue.

● **General Comments:**

1. *The aerosol inlet system on the aircraft should be discussed in the manuscript. It strong related to how to evaluate the accuracy of airborne aerosol measurement.*

**Reply:** Thanks for your comment. We have added some detailed description about aerosol inlet system in Section 2.2 as "The conical double diffuser aerosol inlet, designed for a Twin Otter, was installed on the Y-12. This inlet described by Hegg et al. (2005) has been used extensively on the UMD Cessna 402 (Brent et al., 2014) manufactured by the Droplet Measurements Technologies (MP-1806-A and MP-1807-A, Boulder, CO). The passing efficiency is expected to be near 100% for particle diameters up to 2.5 μm and near 50% for particles between 3-4 μm (Huebert et al., 2004; McNaughton et al., 2007)."

2. *SSA is a very sensitive and unstable parameter. Authors should provide the detail descriptions on how to calculate the columnar SSA from airborne measurement data and their uncertainties? When different aerosol types or multi aerosol layer in vertical, how authors apply to the calculation?*

**Reply:** Thanks for your comment. We have added some detailed description about the calculation of columnar SSA in Section 3.1 as "The method we calculated the columnar SSA from airborne measurement data and estimation of its uncertainties were based on previous studies (Leahy et al., 2007;Schafer et al., 2014). The SSA ($\omega_{sample}$) was calculated from $\sigma_{sca}$ measurements at 450, 550, and 700 nm measured by a Nephelometer and $\sigma_{abs}$ measurements at 565nm by a PSAP. The small mismatch in the wavelength around 550nm was corrected by linearly extrapolating the $\sigma_{abs}$ values. We assumed that *in-situ* SSA measured between minimum and maximum flight altitudes represents the entire column. In order to compare a column SSA ($\omega_{column}$) value with AERONET measurements, the sampled SSA values were averaged for duration of the profile sampling after weighting the values according to aerosol loading. In this study, aerosol loading could represent by $\sigma_{sca}$ values and the profiles were limited to those samples collected from below 400 m AGL and continued to greater than 2000 m that may adequately represented the column value. For every profile, SSA data was weighted by the normalized magnitude of $\sigma_{sca}$ using the following equation as given below:

$$\omega_{column} = \frac{\sum_{i=0}^{N}\left[\frac{\sigma_{sca}}{\sigma_{sca(profile\_mean)}} * \omega_{sample}\right]}{N} \qquad (1)$$

N equals the number of $\omega_{sample}$ in the profile."

Considering the vertical distribution of $\sigma_{sca}$ and $\sigma_{abs}$, the SSA measured at higher altitudes (lower aerosol loading) is substantially less than in the lower troposphere or the aerosol enrichment layer. Thus, the weighted mean method was better than a simple average of $\omega_{sample}$, which would over-represent the absorption features of aerosol that has negligible effect on radiation at the surface where the CIMEL radiometer is located.

3. *Expending the discussion on the Table 3. Why only 28 May, 2016 case shown in the table? Maybe the scatter plot of AERONET SSA vs Aircraft in situ for each flight can provide more strong conclusion.*

**Reply:** The discussion of Table 3 has been expended in the revised manuscript as the following: "These SSA values obtained in the NCP are lower than those observed in Africa and in northeast United States. The reason for the difference is probably due to different type of aerosol components in these different locations: primarily mineral dust aerosols in Africa and photochemically produced secondary aerosols in Northeast US, and the mixed of these two types of aerosols in the NCP."

Due to weather conditions, the restriction of airspace, and the working status of CIMEL radiometer, it was very difficult in acquiring coincident surface and aircraft measurements, even with multiple flights on 10 different days. By given the thresholds used for temporal and spatial matching during the experiment period, only the case in 28 May matched the time and space qualification.

4. *How to calculate AOD for different layers (Fig 5) should describe in the manuscript.*

**Reply:** We have included the following description in Section 3.2 in the revised manuscript:
"During the experimental period, the majority of aerosol layers were well characterized by the sampled vertical profiles and most aerosols reside below the maximum flight levels. Mie theory was applied to calculate the extinction profiles and the AOD, and to estimate the impact of different aerosol vertical distributions on these optical properties. The AOD can be calculated by integrating the extinction coefficient over height as:

$$AOD_{(z1\sim z2)} = \int_{Z_1}^{Z_2} \sigma_{ext}(z)dz \ , \qquad (7)$$

Where the $\sigma_{sca}(z)$ is the extinction coefficient at a height of z and $Z_2$ is above most of the aerosol."

5. *In figure 5(b), it is not clear how authors define the PBL height? Although in the manuscript, the authors say they determined by aerosol scatter profile (Page 9, line 28), does that quantitatively define PBL height? In figure 5(b), how does*

*authors apply the normalization? Does use 1400m PBL height or individual PBL height from each profiles?*

**Reply:** Thanks for your comment. We have added some detailed description about PBL height in Section 4.2 as "The PBL height is determined by the shapes of $\sigma_{sca}$ vertical profiles. When the pollution in the lower layer of the troposphere, the magnitude of $\sigma_{sca}$ increased slightly with height until a layer where $\sigma_{sca}$ sharply decreased. In this study, the mean decreasing rate is about 0.81 $Mm^{-1}\,m^{-1}$, We defined the bottom of this layer as PBL height ($H_{PBL}$)."

We did the normalization by the shapes of $\sigma_{sca}$ vertical profiles. Considering the $\sigma_{sca}$ gradient, we use individual PBL height from each profile. The $H_{PBL}$ calculated (Section 4.2) in the manuscript is shown in Fig. 1 below, and the mean value of $H_{PBL}$ is ~1400m.

[Figure]

**Fig.1 The calculated PBL height of each profile based on airborne observations. The mean value of $H_{PBL}$ is ~1400 m during the experimental period.**

**6. *It is not clear why the authors use the scale height (eq 6 and 7) here? Is that only to calculate the mean profile (or Fit mean in Fig 6)?***

**Reply:** The scale height ($H_p$) represents the height when the aerosol is reduced to 1/e of its surface value. We use $H_p$ as one of parameters to describe a parameterized model of $\sigma_{sca}$ distribution. $H_p$ is determined from airborne observations. It is only calculated in the mean profile of $\sigma_{sca}$ in Fig 6.

**7. *RH and scattering coefficient have strong correlation. Does mean that the PBL aerosol is well mixed and the different in aerosol scatter is due to aerosol hygroscopcity?***

**Reply:** There is a high correlation between RH and $\sigma_{sca}$ under relative dry conditions (both in clean PBL and lower polluted PBL, $r^2$=0.62 and 0.95, respectively), but not under humid conditions.

We have modified Fig. 6b & 7b in the revised manuscript by separating dry and humid conditions and improved the equations correspondingly. In the clean PBL, $H_{RS}$ was proposed by representing a relative stable layer near surface where the vertical variation of $\sigma_{sca}$ was not significant. The mean $\sigma_{sca}$ under dry condition are slightly

lower than under humid condition in this layer, the difference of $\sigma_{sca}$ maybe due to aerosol hygroscopicity. The same can be inferred that the magnitude of $\sigma_{sca}$ increased slightly with height in the polluted lower PBL (Fig.2b below), suggesting that aerosol pollutants were well mixed and the difference of $\sigma_{sca}$ may be due to aerosol hygroscopicity.

[Figure]

**Fig. 2. Mean vertical distributions of $\sigma_{sca}$ at 550 nm (in Mm$^{-1}$) and relative humidity (%) during the flight campaign for those cases of (a, c) clean PBL and (b, d) pollution in the lower layer of the PBL where PBL heights have been normalized to the same altitude. Grey dashed lines represent mean $\sigma_{sca}$ vertical profiles, the light pink and blue dots represent 1s Nephelometer-measured $\sigma_{sca}$, under dry of humid condition respectively. Thick lines show the calculated fitting curves of the $\sigma_{sca}$ profiles (see Eq. 6 and 7). Magenta and blue lines represent RH data collected under dry or humid conditions (c, d). The horizontal error bars represent the standard deviations at every 100 m level.**

- **Specific Comments:**

1. *Figure 2a: why the large discrepancy between airborne and sounding data? In the airborne T profile, one can not easy determine the inversion layer.*

**Reply:** The difference between airborne and sounding data could be due to the mismatch of temporal and spatial measurements. Due to the considerations of flight safety and the restricted control of airspace, the research aircraft and the sounding balloons were rarely collocated. In Fig. 2 of the manuscript, there is about 1-1.5 hours interval between aircraft and sounding measured T profiles.

The T profile in Fig. 2 of the revised manuscript is the average of 3 spirals shown in the following figure. Two inversion layers are identified by inspection at ~1000 m and ~2500 m, which matched well with sounding profile. In the T profile in the manuscript, it is hard to determine the inversion layer probably due to the averaged calculation and smoothed curve.

[Figure]

**Fig. 3 Vertical profiles of temperature (T) from radiosonde and aircraft measurements made on 8 May 2016 over the Xingtai supersite. Each colored line represents the measurement in one spiral.**

**2.  *Equation (1): what is the r value?**

**Reply:** The $\gamma$ is an experimentally determined variable of the hygroscopicity, with water uptake increasing with increasing $\gamma$. Due to the generally hygroscopic nature of aerosol, there is a change in the scattering coefficient measurement. We carried out a calibration to provide a stable scattering data under humid condition. The $\gamma$ was determined to be 0.33. The scattering values are adjusted using a correction factor f(RH). Details of Equation (1) can be referred the study of Shinozuka et al. (2007) and Beyersdorf et al. (2016).

**3.  *Equation (2): what is the C value? B is beta right?**

**Reply:** C is the angular truncation factor. When using the nephelometer for extinction budget studies, correction factors should be applied to account for the effects of angular nonidealities (primarily, the truncation of near-forward scattering). Actual range of total scattering angles captured by nephelometer less than ideal range of 0° to 180°. C was empirically derived by angstrom exponent as Table 1. Details of Equation (2) can be referred the study of Anderson and Ogren (1998).

**Table 1 Parameter adjustments with empirically derived angular truncation correction factors**

| $\lambda$ | Angular Truncation Factor | Angstrom Exponent | Detection Limit |
|---|---|---|---|
| 450 nm | $C^{450} = 1.165 - 0.046 \times A_{550}$ | $A_{550} = -\log(b_{scat}^{450}/b_{scat}^{550})/\log(450/550)$ | 4.4E-07 m$^{-1}$ |
| 550 nm | $C^{550} = 1.152 - 0.044 \times A_{575}$ | $A_{575} = -\log(b_{scat}^{450}/b_{scat}^{700})/\log(450/700)$ | 1.7E-07 m$^{-1}$ |
| 700 nm | $C^{700} = 1.120 - 0.035 \times A_{615}$ | $A_{615} = -\log(b_{scat}^{550}/b_{scat}^{700})/\log(550/700)$ | 1.6E-07 m$^{-1}$ |

$B_{sca}$ is measured scattering coefficient, and $B_{sca\_adj}$ is the adjusted scattering coefficient (corresponding to $\sigma_{sca}$) in the following analysis.

4. ***Does adjusted Beta sca (Beta sca_adj) use in the follow analysis? If yes, author should maintain the symbol consistency.***

**Reply:** $B_{sca\_adj}$ is the adjusted scattering coefficient (corresponding to $\sigma_{sca}$) in the following analysis. Thanks for your comment. We have revised this to maintain the symbol consistency in the manuscript.

5. ***Page 5 line 4: for loading? (not clear)***

**Reply:** "for loading" has been deleted.

6. ***The equation should better insert in a paragraph, not like equations(4) and (5).***

**Reply:** We have revised this as suggested:

"The measured SSA values were scaled proportionally to the aerosol loading at the altitude of the observation as in the following equation:

$$\omega_{column} = \frac{\sum_{i=0}^{N}\left[\frac{\sigma_{sca}}{\sigma_{sca(profile\_mean)}}*\omega_{sample}\right]}{N} , \qquad (4)$$

where N equals the number of $\omega_{sample}$ in the profile." and "Based on measurements of aerosol scattering and absorption coefficients, aerosol extinction coefficient ($\sigma_{ext}$, defined as the sum of $\sigma_{sca}$ and $\sigma_{abs}$) and SSA ($\omega$, defined as the ratio of $\sigma_{sca}$ to $\sigma_{ext}$) were calculated at 550 nm following Eq. 5,

$$\omega_{\lambda} = \frac{\sigma_{sca,\lambda}}{\sigma_{ext,\lambda}} = \frac{\sigma_{sca,\lambda}}{\sigma_{sca,\lambda}+\sigma_{abs,\lambda}}. \qquad (5)"$$

7. ***Equation (6): what is Hp value used?***

**Reply:** We use scale height ($H_p$) as one of parameters to describe a parameterized model of $\sigma_{sca}$ distribution. $H_p$ is determined from airborne observations. We have added the following description: "…and $H_p$ is the aerosol scale height ($H_p$ represents the height when $\sigma_{sca}$ is reduced to 1/e of its surface value); $H_{RS}$ represent a relative stable layer near surface where the vertical variation of $\sigma_{sca}$ was not significant. In the cases of clean PBL, $\sigma_{sca,0}$=124 Mm$^{-1}$, $H_p$ =1146 m and $H_{RS}$ =837 m."

8. ***Equation (7): What is k value?***

**Reply:** In Section 4.2 of the manuscript, Equation (7) is now Equation (9) and we have defined k as the increasing rate of $\sigma_{sca}$ in the PBL with a mean value of 0.03 Mm$^{-1}$ m$^{-1}$ in this study.

9. ***Page 10 (lines 5-6): The correlation........is suitable, please remove since not necessary (not relevant) to discuss here.***

**Reply:** Thanks for your comment. We have revised this as "Under dry condition, there was a pronounced correlation ($r^2$=0.95) between RH_dry and $\sigma_{sca}$ profiles. But under humid condition, the correlation coefficient was 0.12, which suggest a poor correlation between RH_humid and $\sigma_{sca}$ profiles.".

10. ***Page 10, line 21: Why affected by the long-range transport?***

**Reply:** We think that the distribution of $\sigma_{sca}$ showed in Section 4.3 might but not necessarily be affected by long-range transport and local emissions. The enrichment of aerosol might have two causes. Aerosols could be lifted into the free troposphere (e.g., dust aerosol) and transported over a long distance, altering the aerosol vertical distributions over remote areas (Han et al., 2008). Topographically-generated local circulations can also carry high concentrations of surface air pollutants and change the PBL structure (Chen et al., 2009). Thus, we correct the expression of the manuscript as following:

"The vertically inhomogeneous distribution of $\sigma_{sca}$ suggests that aerosol particles in the PBL might be significantly affected by the long-range transport of air pollutants or local emissions around the study area."

**11. Page 10, line 23: Figure 9c, 9d.**
Reply: Thanks for your comment. This has been corrected in the revised manuscript.

**References:**

Anderson, T. L., and Ogren, J. A.: Determining Aerosol Radiative Properties Using the TSI 3563 Integrating Nephelometer, Aerosol Science and Technology, 29, 57-69, 1998.

Beyersdorf, A. J., Ziemba, L. D., Chen, G., Corr, C. A., Crawford, J. H., Diskin, G. S., Moore, R. H., Thornhill, K. L., Winstead, E. L., and Anderson, B. E.: The impacts of aerosol loading, composition, and water uptake on aerosol extinction variability in the Baltimore-Washington, D.C. region, Atmospheric Chemistry & Physics, 16, 1003-1015, 2016.

Leahy, L. V., Anderson, T. L., Eck, T. F., and Bergstrom, R. W.: A synthesis of single scattering albedo of biomass burning aerosol over southern Africa during SAFARI 2000, Geophysical Research Letters, 34, 261-263, 2007.

Schafer, J. S., Eck, T. F., Holben, B. N., Thornhill, K. L., Anderson, B. E., Sinyuk, A., Giles, D. M., Winstead, E. L., Ziemba, L. D., and Beyersdorf, A. J.: Intercomparison of aerosol single‐scattering albedo derived from AERONET surface radiometers and LARGE in situ aircraft profiles during the 2011 DRAGON‐MD and DISCOVER‐AQ experiments, Journal of Geophysical Research Atmospheres, 119, 7439–7452, 2014.

Shinozuka, Y., Clarke, A. D., Howell, S. G., Kapustin, V. N., Mcnaughton, C. S., Zhou, J., and Anderson, B. E.: Aircraft profiles of aerosol microphysics and optical properties over North America: Aerosol optical depth and its association with PM2.5 and water uptake, Journal of Geophysical Research Atmospheres, 112, 1037-1044, 2007.